# Recent Advances in Broadband Photodetectors from Infrared to Terahertz

**DOI:** 10.3390/mi15040427

**Published:** 2024-03-22

**Authors:** Wei Si, Wenbin Zhou, Xiangze Liu, Ke Wang, Yiming Liao, Feng Yan, Xiaoli Ji

**Affiliations:** 1School of Electronic Science and Engineering, Nanjing University, Nanjing 210023, China; 2School of Electronic and Optical Engineering, Nanjing University of Science and Technology, Nanjing 210094, China

**Keywords:** broadband photodetectors, infrared, terahertz, photoelectric detectors, thermoelectric detectors

## Abstract

The growing need for the multiband photodetection of a single scene has promoted the development of both multispectral coupling and broadband detection technologies. Photodetectors operating across the infrared (IR) to terahertz (THz) regions have many applications such as in optical communications, sensing imaging, material identification, and biomedical detection. In this review, we present a comprehensive overview of the latest advances in broadband photodetectors operating in the infrared to terahertz range, highlighting their classification, operating principles, and performance characteristics. We discuss the challenges faced in achieving broadband detection and summarize various strategies employed to extend the spectral response of photodetectors. Lastly, we conclude by outlining future research directions in the field of broadband photodetection, including the utilization of novel materials, artificial microstructure, and integration schemes to overcome current limitations. These innovative methodologies have the potential to achieve high-performance, ultra-broadband photodetectors.

## 1. Introduction

Electromagnetic waves span a broad spectrum, encompassing gamma rays to radio waves, with each segment holding immense value for humanity. Photodetectors serve as a pivotal tool in converting the optical signals of electromagnetic waves into electrical signals, unlocking a plethora of information and finding crucial applications across various modern domains [1,2,3,4,5]. For instance, photodetectors operating in the ultraviolet (UV) spectrum are being widely used in industry and medicine [6,7]. The detection of visible light (VIS) forms the bedrock of contemporary digital camera technology [8]. Within the electromagnetic spectrum, the adjacent bands of infrared (IR) and terahertz (THz) occupy significant positions. As shown in Figure 1, IR and THz detectors play indispensable roles in major fields [9,10]. Infrared photoelectric detectors are well known to the public due to the wide application of infrared thermometers. In addition to its medical value, infrared detection technology has wide and deep applications in military and civilian fields such as reconnaissance, night vision, security, and environmental monitoring [11,12,13]. Extending to the terahertz region, terahertz light (0.1–10 THz) has significant value in security and nondestructive testing applications due to its low photon energy. The distinctive advantages of terahertz spectroscopy in characterizing absorption make it particularly suitable for the study of large biomolecules, and it could be applied to achieve super-resolution biomedical imaging. Furthermore, terahertz photodetectors are pivotal in high-speed communications and astronomical remote sensing [14,15,16,17]. Particularly, the terahertz band is a unique frequency band for detecting the early cold universe, the dark universe, and the environment of life in the universe [18].

In recent years, the development of superior optoelectronic materials and innovative sensor structures has promoted advancements in both infrared and terahertz detector fields [19,20,21,22,23,24], and some have even been commercialized [25,26,27,28,29,30]. However, the expanding range of wavelengths that need to be detected makes conventional single infrared or terahertz photodetectors inadequate for meeting the demands of increasingly sophisticated optoelectronic sensing [31,32]. Consequently, there has been a surge of interest in the development of high-performance broadband photodetectors. The Lager Ultraviolet/Optical/Infrared Surveyor (LUVOIR) [33] space observatory developed by NASA in the United States will be the first one to detect exoplanets that are the most similar to the Earth; LUVOIR will also seek signs of life within the solar system and beyond. In an international collaboration, China and France are working on a dedicated satellite, Space-Based Multi-Band Astronomical Variable Objects Monitor (SVOM) [34,35], designed to provide rapid multiband observations of high-energy, transient astronomical events such as gamma-ray bursts (GRBs) and afterglows. Remote sensing researchers in Germany have proposed the multispectral Dynamic Infrared Earth Observation on the ISS Orbit (DIEGO) [36] sensor, with 11 spectral bands for narrowing the observational gap between LWIR and MWIR. The data it provides will be instrumental in various fields, including meteorology, climatology, and environmental science. In addition to the above applications, SiGe-based broadband visible–NIR imagers have provided low-cost solutions for a wide range of military applications such as day–night vision, biochemical threat detection, muzzle flash detection, and more [37]. The U.S. Air Force Office of Scientific Research has helped researchers develop dynamic multiband image datasets for image fusion in military and civilian short-range surveillance scenarios [38]. The infrared to terahertz spectral range, from a few micrometers to several hundred micrometers, is a key region for bridging the gap between conventional infrared and microwave technologies. The realization of high-performance infrared to terahertz broadband photodetectors would not only enhance established applications in astronomy and military technology, but also unlock novel opportunities for applications such as optical communications, spectroscopy, medical imaging, environmental monitoring, etc., providing insights into previously inaccessible material properties, chemical compositions, and biological phenomena [39,40,41,42]. Researchers have conducted reviews on the progress of photodetectors related to infrared and terahertz topics. However, these reviews either exclusively summarized broadband detectors based on single-type materials (e.g., two-dimensional materials) [3] or focused solely on research advancements in infrared or terahertz wavelength, which cannot work in both bands [43]. There is a lack of reviews covering various types of photodetectors operating across the infrared to terahertz regions.

In this review, we provide a comprehensive overview of the latest advancements, challenges, and prospects in the field of infrared to terahertz photodetectors. We delve into device classifications, fundamentals, emerging materials, and process optimization in the field, and we summarize the diverse strategies employed to extend the spectral response of photodetectors that have catalyzed transformative progress. Furthermore, we discuss prevailing challenges in the development of broadband photodetectors and outline potential future research directions. These challenges encapsulate the pursuit of higher performance, miniaturization, cost-effectiveness, and integration with complementary technologies. We also highlight emerging trends in materials research, innovative micro- and nanostructures, and device integration schemes that hold great promise for realizing the next generation of infrared/terahertz detectors.

## 2. Fundamentals of Photodetectors

Photodetectors convert optical signals to electrical signals through light absorption, during which extra charge carriers are produced [44,45]. In this section, various detection mechanisms are summarized, along with the key characteristics that affect the performance of photodetectors.

### 2.1. Detection Mechanism

Typically, there are two primary types of photodetection mechanisms: (1) thermal effects, which generate free charge carriers and give rise to photodetection, including the photo-bolometric effect (PBE) and the photo-thermoelectric effect (PTE); (2) light incident on the active channel of the device, which excites free charge carriers by optical transition, including the photoconductive effect (PCE), photovoltaic effect (PVE), and photogating effect (PGE). In addition to the common photodetection principles, many mechanisms based on other physical principles have also been discovered in recent years, such as thermal phase transition, pyroelectric, and electromagnetic-induced wells (EIW), as well as devices that combine multiple detection mechanisms [3]. Figure 2 below provides a schematic of these mechanisms, where the “Others” box illustrate the novel EIW effect.

#### 2.1.1. Photo-Bolometric Effect (PBE)

The photo-bolometric effect refers to the changes in the resistivity or conductivity of a thermal-sensitive material due to the absorption of photons that cause a uniform optical heating effect. As shown in Figure 2, when light is incident on a thermally sensitive material, the material is subject to a temperature change (Δ*T*) and the following changes in the electrical properties, which could be manifested as either an increase or a decrease in the resistance of the material (Δ*R*). The current across the device could also increase or decrease at a fixed bias voltage. The photocurrent governed by the photo-bolometric effect increases with a linear increase in the bias voltage. The key performance parameter of the photo-bolometric effect is the thermal resistance (*R_h_*) [3]:(1)$$R_{h}=\frac{dT}{dP}$$
where *dT* is the temperature at which the device is elevated; *dP* is the absorbed incident radiation power. The thermal resistance ultimately determines the sensitivity of the device. Consider the thermal capacitance *C_h_*, which determines the response time of the device: *τ* = *R_h_C_h_*. PBE is often observed in materials with strong electron–phonon interactions, where the absorbed light energy leads to an increase in lattice vibrations and subsequent changes in conductivity.

#### 2.1.2. Photo-Thermoelectric Effect (PTE)

The photo-thermal effect (Seebeck effect) is a thermoelectric effect that generates a voltage or current within a material due to uneven heating caused by incident light. When the spot is smaller than the dimensions of the device channel, localized light irradiation creates a temperature gradient in the semiconductor channel and leads to a temperature difference Δ*T* in different parts or ends of the device. As shown in Figure 2, the Δ*T* leads to the generation of a potential difference. The Seebeck effect is the effect that describes the generation of voltage in a conductor when a temperature gradient exists. The thermoelectric voltage (*V_PTE_*) can be measured as an electrical output and can be represented as [46]
(2)$$V_{\mathrm{PTE}}=(S_{1}-S_{2})\Delta T$$
where *S_1_* and *S_2_* are the Seebeck coefficients of the material in units of V K^−1^. The most important distinction between a PTE and a PBE is that the photocurrent generated by a PTE does not need to be driven by an external voltage, whereas the photocurrent cannot be observed in a PBE without applying an external bias. In addition, the sign of the photoresponse is different. For the PBE, the sign of the photocurrent depends on the change in resistivity, which is a function of temperature. For the PTE, it depends on the difference in Seebeck coefficients between the material regions in the semiconductor and the carrier polarity.

#### 2.1.3. Photoconductive Effect (PCE)

The photoconductive effect is the effect of the light-induced generation of excess carriers, leading to an increase in the concentration of free carriers and an increase in conductivity in a semiconductor material. In the absence of light, a small dark current (*I_dark_*) flows through the device under the application of a biased voltage *V_dS_*. When there is light (energy of photons (*E_p_*) *>* bandgap energy of the material (*E_g_*)) irradiating the photoconductive material, the material absorbs incident photons with energies greater than or equal to the bandgap energy and thereby generates electron–hole pairs. These charge carriers increase the conductivity of the material. The photogenerated excess carrier density Δ*n* increases the conductance by Δ*σ =* Δ*nqμ* [1], where *μ* is the mobility of the channel materials. Upon applying a bias voltage, these carriers are separated and driven, resulting in a light current (*I_light_*) that is greater than the dark current, as shown in Figure 2. The photocurrent (*I_P_*) can be expressed as [1]
(3)$$I_{P}=I_{\mathrm{light}}-I_{\mathrm{dark}}$$

Typically, photoconductive materials are semiconductors whose conductivity can be modulated by the intensity of incident light. Specifically, photoconductive devices require the application of an external bias voltage during detection to separate light-generated carriers (electron–hole pairs) to form a photocurrent.

#### 2.1.4. Photovoltaic Effect (PVE)

The photovoltaic effect is the effect in which the built-in electric field of a device separates the photogenerated carriers rapidly and generates a photogenerated electric field in the opposite direction to the built-in electric field. The photovoltaic effect can be realized by using two different materials with different figures of merit. As shown in Figure 2, when light radiates the p–n junction or Schottky junction formed at the contact interface between a semiconductor and a metal, the p–n junction or Schottky junction absorbs photons and generates photogenerated carriers. Then, these carriers are separated spatially by the built-in electric field, leading to the aggregation of photogenerated electrons near the boundary of the n region, and photogenerated holes near the p-region boundary. This configuration creates a photogenerated electric field in the opposite direction to the built-in electric field. In the presence of light radiation, a current could be collected by connecting electrodes to the device, thereby converting light energy into electrical energy. Photodetectors based on the photovoltaic effect (PVE) typically operate self-powered, obviating the need for an external bias voltage.

#### 2.1.5. Photogating Effect (PGE)

The photogating effect is a special mechanism of the photoconductive effect, which refers to the modulation of electrical properties such as resistance or conductance in a material by incident light. Typically, the photogating effect is present in semiconductor devices with heterogeneous structures or defects. Upon irradiation of the device with light, free electron–hole pairs are generated. Localized states of the semiconductor device could trap specific kinds of charge carriers, and the charged trap states act as localized floating gates, which strongly modulate the channel conductance through electrostatic interactions, as shown in Figure 2. Thus, the conductivity can be modulated effectively. For example, hole-trapped states are positively charged after the photogenerated holes are trapped, which leads to additional free electrons in the device. In this case, the transfer curve demonstrates a horizontal transfer in the negative direction (Δ*V_g_*). In the PGE process, the lifetime of the photogenerated carriers becomes longer due to the capture of opposite charge carriers, which results in high gain. Based on the conductivity Δ*σ =* Δ*nqμ*, we note that both the photoconductive and photogating effects modify the conductivity by *n*. The PGE can be used in optical switching and photodetection applications.

In summary, these effects demonstrate the various ways in which light interacts with materials, leading to changes in electrical properties. The photo-bolometric effect, photo-thermal effect, photoconductive effect, photovoltaic effect, and photogating effect all play important roles in the development of light detection, energy conversion, and optoelectronic devices. The emerging EIW mechanism is the photovoltaic effect, which is not limited by the material bandgap. Based on this mechanism, it is possible to broaden the response range of detectors; additionally, this mechanism shows promise for meeting the future requirements of high-performance detection at room temperature.

### 2.2. Key Characteristics of Photodetectors

To better compare the performance of photodetectors with different principles, sizes, and operating conditions, we summarized a set of device performance metrics. The following section introduces these key characteristics briefly.

#### 2.2.1. Responsivity (*R*)

Responsivity is a fundamental parameter that characterizes the sensitivity of a photodetector. It quantifies the relationship between the optical power incident on the detector and the resulting electrical output signal. Responsivity is expressed as the ratio of the output electrical current or voltage to the incident optical power, typically measured in volts per watt (V W^−1^) or amps per watt (A W^−1^). Mathematically, the responsivity (*R*) of a photodetector is given by [1]
(4)$$R_{V}=\frac{V_{p}}{P_{\mathrm{in}}}\;\mathrm{or}\;R_{I}=\frac{I_{p}}{P_{\mathrm{in}}}$$
where *R_V_* or *R_I_* is the responsivity in V W^−1^ or A W^−1^, *V_p_* or *I_p_* is the output voltage or current generated by the photodetector, and *P_in_* is the incident optical power in watts.

It is important to note that responsivity is typically specified at a specific wavelength or wavelength range for a given photodetector. Different photodetectors have different responsivity characteristics, and the choice of photodetector depends on the application’s wavelength requirements.

#### 2.2.2. Noise Equivalent Power (NEP)

The noise-equivalent power (NEP) is a critical parameter that characterizes the sensitivity of a photodetector. It represents the minimum detectable optical power that can be distinguished from the noise in the electrical output signal of a detector. In other words, NEP is a measure of a detector’s ability to detect weak optical signals in the presence of noise, and it indicates the photodetector’s signal-to-noise performance.

NEP is typically specified in units of watts per square root of hertz (W Hz^−1/2^). The lower the NEP value, the more sensitive the photodetector to small optical power variations, enabling it to detect weaker signals. Mathematically, the NEP is given by [3]
(5)$$\mathrm{NEP}=\frac{V_{n}}{R_{v}}=\frac{I_{n}}{R_{I}}$$
where *V_n_* or *I_n_* is the electrical noise present in the output signal of the photodetector, usually measured in volts (V) or amperes (A). *R_V_* or *R_I_* is the sensitivity of the photodetector, representing the electrical output generated per unit of incident optical power (V W^−1^ or A W^−1^). It is important to note that NEP is generally specified at a specific wavelength or wavelength range, as it is influenced by the photodetector’s responsivity at that particular wavelength.

The NEP is closely related to the noise figure (NF) of a photodetector system. The noise figure is a measure of how much the noise from the photodetector and the subsequent electronic components in the system degrades the overall signal-to-noise ratio of the system. The lower the NEP and NF, the better the photodetector’s performance in detecting weak optical signals with high precision.

#### 2.2.3. Detectivity (*D**)

Detectivity is also a critical figure of merit of a photodetector and provides a quantitative measure of its ability to detect weak optical signals in the presence of noise. It is particularly useful when comparing the performance of different photodetectors for specific applications where high sensitivity is crucial. The detectivity *D* is the reciprocal of the NEP [3]:(6)$$D=\frac{1}{\mathrm{NEP}}$$

Jones [4] found that for many detectors, the NEP is proportional to the square root of the detector signal, which is proportional to the detector area *A_d_*. This implies that the NEP and the detection rate are functions of the detector area and the electrical bandwidth, hence the normalized detection rate *D** (or D-star) proposed by Jones. *D** is expressed in units of Jones (cm Hz^1/2^ W^−1^). The higher the detectivity value, the more sensitive the photodetector, enabling it to detect smaller amounts of light with high precision. Mathematically, the detectivity (*D**) is given by [3]
(7)$$D*=\frac{\sqrt{A_{d}\Delta f}}{\mathrm{NEP}}$$
where *A_d_* is the area of the device, and Δ*f* is the electrical bandwidth of the device.

Detectivity is a comprehensive parameter that combines the photodetector’s responsivity and noise performance into a single metric. It is widely used to compare the sensitivity of different photodetectors, regardless of their specific size or active area.

#### 2.2.4. Quantum Efficiency

The external quantum efficiency (EQE) serves as an index that measures the conversion efficiency of photoelectric devices for incident photons under actual operating conditions. It elucidates the relationship between the output current of a photodetector and the number of incident photons. EQE is typically determined by measuring the device’s output current response, with the following formula [1]:(8)$$\mathrm{EQE}=\frac{N_{c}}{N_{I}}=\frac{\mathrm{hc}}{e\lambda }R$$
where *N_C_* represents the number of carriers collected, and *N_I_* represents the number of photons illuminating the device. *h* is the Planck constant, *c* is the speed of light, *e* is the electron charge, and *λ* is the wavelength of the incident light. The higher the EQE value, the higher the conversion efficiency of the device. Upon illumination, only a portion of the photons are absorbed. The quantity of absorbed photons, denoted as *N_A_*, can be expressed as *N_A_* = *N_I_ η_A_*, where *η_A_* represents the light absorption efficiency. In contrast, internal quantum efficiency (IQE) focuses on the efficiency of converting photons into charge carriers after being absorbed inside the device. Its formula is [1]
(9)$$\mathrm{IQE}=\frac{N_{c}}{N_{A}}=\frac{\mathrm{EQE}}{\eta_{A}}$$

By enhancing the IQE, the internal photoelectric conversion efficiency of a device can be improved. These two parameters are crucial for evaluating the performance of photoelectric devices. EQE provides information on the conversion efficiency of the entire device to incident photons, while IQE delves deeper into the efficiency of the photoelectric conversion process occurring within the device itself.

#### 2.2.5. Response Time

The response time of a photodetector is the rate at which a detector produces an electrical output signal in response to a change in incident light intensity, which quantifies how quickly the photodetector responds to changes in light. The response time is usually expressed in terms of rise time (*τ_r_*) or fall time (*τ_f_*). In terms of rise time, it is the time it takes for the output net photocurrent of a photodetector to rise from some low percentage (e.g., 10%) of its maximum value to a higher percentage (e.g., 90%) when exposed to a sudden increase in light intensity. The fall time, on the other hand, is the time required to fall from 90% to 10%. The current (*I*) during device operation can be expressed by the following equation [46]:(10)$$I={\;I}_{\mathrm{pho}}e^{-(\frac{t}{\tau })\beta }$$
where *I_pho_* represents the current in the device in the absence of light exposure, *β* is the exponential factor, and *τ* is the response time. Typically, the response time is in ms or µs. The response time is affected by factors such as the detection mechanism, the effective area of the detector, the material properties, and the readout circuitry. For example, for photoconductive devices, the response time is related to carrier mobility, channel length, and applied bias voltage. For photovoltaic devices, the response time is related to the thickness of the active area of the detector and carrier mobility. Response time must be considered when selecting a photodetector for a particular application. Some applications may require a balance between sensitivity and response time, as faster detectors may sacrifice sensitivity to achieve their speed. The best choice depends on the specific requirements of the application scenario and the trade-off between speed and sensitivity.

#### 2.2.6. Spectral Response Range

The spectral response range of a photodetector is the range of wavelengths over which the detector is sensitive to and capable of producing a photoelectric response. Different photodetector technologies have unique spectral response characteristics that make them suitable for specific applications based on the desired wavelength range. For photonic detectors, the spectral response range is limited by the forbidden bandwidth. Silicon photodiodes are typically used in the visible and near-infrared regions, while specialized detectors such as indium gallium arsenide (InGaAs) photodiodes are suitable for the near-infrared region. Avalanche photodiodes (APDs) and superconducting detectors can extend sensitivity to higher wavelengths, including the mid-infrared and terahertz ranges [43,44]. For thermal detectors, a larger spectral response is generally possible. Understanding the range of spectral response is critical in selecting the right photodetector for a particular application.

## 3. Broadband Photodetectors Based on Thermal Effects

Broadband photodetectors based on thermal effects (e.g., bolometric effect, photo thermoelectric effect, pyroelectric effect, and thermal phase transition) utilize the absorption of light energy to generate heat, which causes a temperature change in the detector material so it can detect the optical signal. This type of detector has no bandgap limitation, does not require specific photon energy to be excited, and can theoretically be used to achieve high-sensitivity optical detection in the ultra-wideband spectrum. It should be noted that different types of thermal effect-based ultra-wideband photodetectors may use different materials and structural designs to achieve the best performance and application range.

### 3.1. PBE-Type Broadband Photodetector

In the mid-infrared to terahertz region, photodetectors based on the photo bolometric effect become the main choice. Micro-bolometric detector technology is used in typical thermal detectors. It is quite compatible with semiconductor manufacturing processes and is suitable for large-scale integration and production. It has become the mainstream technology direction for uncooled IR/THz detectors. At present, the disadvantages of bolometric photodetectors are low detectivity and slow response speed. The detectivity is mainly limited by the temperature coefficient of resistance (TCR) of the material. The TCR in commercial devices is about −2 to −3%/K, which must be improved. A major focus of future research is the development of new materials with high sensitivity and broadband response properties. The response speed of a detector is largely limited by the thermal capacity and thermal conductivity, so researchers are also working to reduce the thermal time constant of devices. Although the TCR and thermal time constant can be improved, these devices are still subject to thermal interference from the thermal noise and 1/f noise. High responsiveness requires large bias voltages, leading to increased power consumption and noise, the presence of which limits the detection capability of bolometric photodetectors.

The broadband photoresponse achieved by PBE-type photodetectors is due to the ultra-broad light-absorbing ability of the sensing material. The material absorbs a broad spectrum of light while having an extremely high thermal conductivity that quickly converts the absorbed energy into uniform heat. Researchers have generally enhanced the photoresponse of such detectors by exploring novel broadband-effect materials and building specific microstructures. Photodetectors based on carbon materials are of great value in realizing the function of ultra-broad detection because of their unique electronic structure and chemical bonding properties, which are characterized by high absorbance. Ji et al. [47] found a negative TCR of about 0.13–0.15%/K in superblack carbon aerogel (CA). They fabricated devices using CA films with a thickness of approximately 400 nm, which exhibited an ultra-broadband response from the UV, visible, and infrared to microwave ranges (300 nm to 3 cm). As shown in Figure 3a, with increasing temperature, electrons first flow out of the energy band of carbon quantum dots and then inject into the conduction band of graphite, which increases the carrier concentration. Owing to the hot-electron effect, the photothermal conversion and the resulting electromagnetic wave response are broadened greatly. This study showed that a broadband optical response can be realized by combining the high photothermal conversion efficiency and TCR properties of a material. Due to its special thermodynamic properties, the graphene in carbon-based materials has also been used as an ultra-sensitive bolometric detector in the far-infrared range at low temperatures. At the same time, the use of plasma structures, such as nanoantennas, can enhance the interaction between light and matter and thus expand the bandwidth of photodetectors. Mittendorff et al. [48] presented a graphene-based ultra-fast photodetector that operates in the infrared and terahertz range (8 μm to 220 μm) at room temperature. As shown in Figure 3b, a broadband response was achieved by machining a logarithmic-periodic antenna on a graphene sheet by electron beam lithography. In addition, the researchers explored the important role of the substrate, using a high-resistance substrate that does not absorb in the target wavelength range, such as diamond or SiC, to broaden the detector wavelength range.

Different from graphene prepared by hand tearing, reduced graphene oxide (RGO) has unique advantages such as a wet synthesis method, high yield, and the ability to assemble large-area films on various substrates. Therefore, researchers have also begun to explore the thermal properties of RGO. To demonstrate the potential of freestanding RGO thin films for optoelectronic device applications, Yang et al. [49] constructed a fully suspended RGO photodetector (Figure 3c) using freestanding RGO thin films in 2017, which exhibited the fastest (~100 ms) and widest (from UV to terahertz spectral ranges) photoresponse among all reported RGO thin-film photodetectors. The response is comparable to that of CVD-grown graphene photodetectors and mechanically exfoliated graphene photodetectors. Carbon nanotube (CNT) thin films are expected to be used in ultra-broadband photodetectors due to their ultra-broad absorption spectrum, high charge carrier mobility, good processability, and high mechanical flexibility. Liu et al. [50] reported an ultra-wideband bolometric photodetector based on suspended carbon nanotube membranes. Figure 3d is a schematic diagram of the device structure and the light response curve under different laser irradiations in air and in a vacuum. With a rich tube diameter distribution and a TCR equal to 0.509%/K, CNT membranes show a strong absorption spectrum from the ultraviolet to the terahertz regions. In the presence of light, the heat generated by electron–photon interactions dominates the optical response of the device. For small temperature changes, the photocurrent shows a convincing linear correlation with the power of absorbed light over three orders of magnitude. The detectivity of the device in the mid-infrared (MIR, 10.6 μm) and terahertz regions (118 μm) is 0.63 × 10^7^ Jones and 2.31 × 10^7^ Jones, respectively.

In addition to new photoresponsive materials such as graphene and carbon nanotubes, vanadium oxide, as a material for commercialized bolometers, has also made some research progress. In recent years, Wadsley-phase vanadium oxide has attracted attention due to its quasi-layered structure, whose properties can be tuned to a two-dimensional layered material. B phase is a typical Wadsley-phase vanadium oxide with a zero bandgap, which is suitable for broadband photodetection at room temperature. Zhang et al. [51] developed B-phase vanadium oxide (VO_2_ (B)) characterized by broadband light absorption, a large TCR, and low noise for uncooled broadband detection. Figure 3e shows the structural diagram of the prepared device, multiband optical response, and terahertz imaging results. By adopting a freestanding structure and reducing the size of the active region, the VO_2_ (B) photodetector exhibited stable and excellent performance in the visible to terahertz region (405 nm to 0.88 mm), with a peak TCR of −4.77%/K at 40 °C, a peak specific detectivity of 6.02 × 10^9^ Jones, and a photoresponse time of 83 ms. The terahertz imaging capability was 30 × 30 pixels. 

In recent years, the continuous development of microbolometers, especially through the rapid popularization of artificial intelligence and the Internet of Things in various fields, not only made microbolometers in the traditional market share surge but also promoted and accelerated the development of microbolometers in the civilian field. To adapt to future applications and markets, the microbolometer development trend will mainly manifest in the following aspects: (1) large array, low cost, high-speed microbolometers; (2) highly integrated microbolometers.

### 3.2. PTE-Type Broadband Photodetector

Photodetectors based on the photo thermoelectric effect (also known as the Seebeck effect) are well suited to meet the need for ultra-wideband response due to their simple device geometry, self-powered capability, low power consumption, and room-temperature operating characteristics, making them suitable for integration into arrays. The PTE effect requires the generation of a large amount of photoelectric voltage through the creation of a temperature gradient via asymmetric illumination of the channel material or localized heating due to an asymmetric device structure. The primary advantage of this mechanism is that thermoelectric conversion is induced by optically generated temperature gradients within the device resulting from different materials. Theoretically, similar to PBE photodetectors, PTE photodetectors exhibit broadband spectral responses, covering the long-wave infrared to terahertz regions, due to the inherent wavelength-insensitive nature of the thermoelectric process. This is achieved by employing materials with an ultra-wideband light-absorption capability. Typical asymmetric device structures include contacts between active material and metal electrodes, dissimilar contact metals, and p–n junctions, of which the first type has the simplest structure and has attracted a lot of research interest. However, the optical response time of photothermal devices is related to thermal dissipation, and, typically, the response speed and responsivity of this type of device is low, comparable to those of ultra-wideband detectors based on the principle of radiometric thermometers, but the noise is significantly less than that of radiometric thermometers. PTE-based devices with millimeter-channel scales have been reported to have slow response times (0.1 s), so there is an urgent need for fast PTE-based photodetectors.

For PTE photodetectors, there are currently two main directions aimed at enhancing their photoresponse. Firstly, the development of photosensitive materials with high absorption rates and Seebeck coefficients has been pursued. Researchers have enhanced the sensitivity of photodetectors by employing materials such as two-dimensional carbon-family materials, EuBiSe_3_ single crystals, topological insulators, and metal halides. The second approach involves increasing the light-induced temperature gradient and the Seebeck coefficient difference within the device. For instance, the plasma-enhanced light absorption strategy has been successfully employed to establish significant temperature gradients.

Two-dimensional carbon-family materials are still being widely studied as photoactive materials in the research on photothermal detectors. The unique gapless band structure of graphene allows the generation of charge carriers via photoexcitation over a very wide spectral range. Meanwhile, the carrier mobility of graphene on the substrate at room temperature is very high, which allows the photons or equipartition excitations to be converted into current or ultra-fast voltage, and the devices can present ultra-fast response times. In 2014, Cai et al. [52] developed a graphene thermoelectric photodetector with a sensitivity of over 10 V W^−1^ at room temperature and a noise equivalent power of less than 1.1 nW Hz^−1/2^, which means that for an optimal coupling device, its performance could compete with the best room-temperature terahertz detector. Time-resolved measurements show that the graphene detectors are eight to nine orders of magnitude faster than those detectors. Despite reported superior performance, monolayer or few-layer graphene exhibits low optical absorption in the mid-infrared range (2.3%). In recent years, researchers have endeavored to enhance its absorption performance through the optimization of graphene’s structure. Li et al. [53] prepared, for the first time, a three-dimensional (3D) graphene foam (GF) diode PD based on an asymmetric Au/3D GF/Ti structure. The excellent light absorption and thermal conductivity enabled nanoporous graphene to be used in broadband energy collectors and detectors. These devices exhibited a high *D** of 10^9^ Jones and a response time of 43 ms. To verify and quantify the photodetector mechanism, the researchers designed a reference device using a symmetric Au/3D GF/Au electrode structure. Figure 4a shows a schematic of the band profiles of these two devices. The results show that the photocurrent is dominated by the PTE effect. Suzuki et al. [54] also reported the direct visualization of heat trapping in the pores of three-dimensional bi-continuous nanoporous graphene membranes and their self-supporting, bendable, broadband infrared (IR) and terahertz (THz) detectors. The detectors produced current changes at 0.14 THz, 1.4 THz, and 39 THz radiation, indicating that the nanoporous graphene devices functioned as broadband THz and IR detectors. Also, the detected signal occurred at zero bias voltage.

Compared to supported graphene-based photodetectors, suspended graphene-based photodetectors have higher photoelectric conversion efficiency, ultra-low 1/f noise, and lower surface plasmon loss. Therefore, the application of multilayer and suspended graphene films in photodetectors has attracted great interest. Wen et al. [55] used freestanding reduced graphene oxide (rGO) films annealed at different temperatures (200–1000 °C) to prepare photodetectors for the first time. The device structure and experimental results are plotted in Figure 4b. The results show that the responsivity of the rGO-based photodetectors decreases by 50% as the annealing temperature increases over an ultrawideband from the ultraviolet (375 nm) to terahertz (118.8 μm) region. This is because the content of electron-donating groups decreases with increasing annealing temperature, which leads to a decrease in the device’s responsivity. Additionally, the device is self-powered and has a response time of <50 ms. Later, Hu et al. [56], from the same group, considered the problem of the ohmic contact between rGO films and electrodes. Metals with high and low figures of merit need to be selected to achieve good p-type doping and n-type doping of rGO films, respectively. At the same time, the selected metals need to exhibit good wettability with rGO and remain stable in atmospheric conditions. They chose palladium and titanium metals to achieve good p-type doping and n-type doping of the films by evaporating palladium and titanium films, respectively. Self-powered suspended Pd-reduced graphene oxide-Ti (Pd-rGO-Ti) photodetectors (Figure 4c) were finally prepared, and the properties of the rGO films were varied by using different annealing temperatures. The resulting detectors exhibited an excellent photoelectric response over a broadband illumination wavelength range of 375 nm to 118.8 μm (2.52 THz).

Carbon nanotubes (CNTs) have been widely used to construct photodetectors based on the PTE effect because of their high thermal conductivity and ultra-broad absorption spectra. Zubair et al. [57] reported the development of a woven, substrate-free, and polarization-sensitive photodetector based on highly aligned carbon nanotubes doped with engineered fibers (Figure 5a). This room-temperature-operated self-powered detector responds to radiation over an ultra-wide spectral range from ultraviolet to terahertz via the photothermal effect. This unconventional photodetector will be applied to wearable technologies that require the detection of electromagnetic radiation. Carbon nanotube forests (CNTFs) are self-aligned, densely arranged, three-dimensional structures of CNTs known for their near-uniform absorption in the ultra-wide infrared range beyond the far infrared. As shown in Figure 5b, Zhang et al. [58] developed a self-powered infrared/terahertz photodetector and energy harvester by exploiting the vertical photo thermoelectric effect of CNTFs. The detector had a broadband detection rate of 1.9 × 10^7^ Jones in the 2.5–25 μm spectral range and a peak detection rate of 2.3 × 10^9^ Jones at 4.3 THz. Owing to the vertical structure, the photodetector exhibited enhanced sensitivity to weak and unfocused IR/THz illumination, which mitigated the high actuation power density found in conventional PTE or field-effect detectors and provides practical IR/THz detection in real-life situations. Similar to rGO, carbon nanotubes also need to have good ohmic contact with electrodes. Liu et al. [59] reported a high-performance ultra-wideband PD with a composite nanostructure consisting of a suspended CNT membrane with titanium and palladium deposited on it, which provides n-doping and p-doping, respectively. Distinguishingly, this study employed numerical simulations to identify that the metal nanoparticles utilized for doping concurrently impart enhanced thermal localization. The device exhibited a response time of ~7 ms, a large specific detectivity (5 × 10^8^ Jones), a large linear dynamic range (30 dB), and small noise-equivalent power (0.05 nW Hz^−1/2^) for ultra-wideband spectroscopy based on strong photothermal effects. Building on this work, Lv et al. [60], from the same group, achieved excellent photothermal and photovoltaic conversion properties by joining silver nanostructured membranes and carbon nanotube membranes through van der Waals forces to form heterojunctions. When silver nanostructures are irradiated with laser light, it induces the enhancement in localized surface plasmon resonances, leading to an enhanced electric field. This electric field subsequently augments the light absorption of the carbon nanotubes. The proposed heterostructures can be used as efficient and sensitive photothermal materials or ultra-wideband fast-response photovoltaic materials. As shown in Figure 5c, when the heterojunction is irradiated with laser light with wavelengths ranging from the ultraviolet to terahertz region, the local temperature difference and output photovoltage increase rapidly. The maximum temperature difference reached 215.9 K, which is significantly higher than that of other photothermal materials reported in the literature for photothermal materials.

In addition to carbon-based materials, transition-metal dichalcogenides (TMDs) have also been a hot research topic for high-performance photodetectors. SnSe_2_-based optoelectronic devices have been developed for the visible and infrared ranges of the electromagnetic spectrum, showing a sharp decrease in efficiency at longer wavelengths. Here, Guo et al. [61] developed SnSe_2_ photodetectors with exfoliated SnSe_2_ nanosheets scaling over the terahertz frequency range, as shown in Figure 5d. The photodetector exhibited high responsivity (170 V W^−1^), fast speed (2.2 μs), and room-temperature operation based on the efficient generation of photovoltaics under deep subwavelength electromagnetic focusing, which outperformed thermal-based photodetectors. The SnSe_2_-based detector showed high-contrast imaging from the terahertz to visible-light ranges. Quasi-one-dimensional (quasi-1D) materials have received little attention except for two-dimensional materials. Due to the large surface-to-volume ratio and small size, quasi-1D materials are suitable for novel flexible optoelectronic devices. In recent studies, quasi-1D transition metal sulfides have shown promise as photosensitive materials. Within this family, NbS_3_ has been almost unnoticed in the field of light detection. Wu et al. [62] reported a PTE detector fabricated from NbS_3_. As shown in Figure 5e, the device exhibits considerable performance in the ultraviolet to terahertz range. The optical response is greater than 1.5 V W^−1^ for all detected wavelengths, and the response time is less than 10 ms, which is much shorter than those reported for ultrawideband photodetectors made of millimeter-scale graphene, ternary chalcogenide single crystal, and other materials. A thorough study of the device principles reveals that the mechanism of the enhanced optical response is the thermally localized enhanced PTE effect.

Recently, rare earth element Eu and bismuth or antimony sulfur compounds (i.e., ternary europium sulfur compounds, EuSbSe_3_, and EuBiSe_3_) have been synthesized based on traditional sulfur–bismuth–antimony bulk alloy thermoelectric materials. These two newly discovered phases have complex crystal structures and large crystalline cells, leading to reduced thermal conductivity. Their very attractive Seebeck coefficients, close to 1000 μV/K at room temperature, offer great potential for thermoelectric applications. As shown in Figure 6a, Wang et al. [63] reported a photothermal detector made of a EuBiSe_3_ single-crystal alloy. The device shows a room-temperature self-powered photoresponse from the ultraviolet (375 nm) to terahertz (163 μm) region with almost uniform sensitivity to wavelength. Due to the large thermoelectric power (Seebeck coefficient) of EuBiSe_3_, the detectivity exceeds 10^8^ Jones across all examined wavelengths, and the noise-equivalent power of the device is less than 1 nW/Hz^−1/2^, which is lower than that of the reported photothermal detectors by one to two orders of magnitude. Three-dimensional topological insulator candidates are promising materials for fast high-performance photodetectors due to their linear dispersion band structure and high carrier mobility. An ultra-broadband photo-thermoelectric (PTE) self-powered detector based on the topological insulator candidate HfTe_5_ was first reported by Niu et al. [64]. As shown in Figure 6b, the photosensitive properties are characterized by an ultra-broadband range from the ultraviolet (375 nm) to terahertz (118.8 µm) wavelengths, and the detectivity was found to be greater than 10^7^ Jones at room temperature for all detected wavelengths. Due to the Dirac energy-band dispersion of HfTe_5_, the proposed detector has a short response time (*τ*) of 1 ms, which is 1–3 orders of magnitude faster than that of the recently reported PTE detectors based on millimeter graphene, 3D graphene, EuBiSe_3_ single crystal, and SrTiO_3_ crystal.

With simple processing methods, ultra-low thermal conductivity, a high Seebeck coefficient, and high carrier mobility, metal halide has unique photoelectric properties and potential PTE characteristics, making it an ideal photoluminescent material for manufacturing flexible, ultra-broadband detectors. The reported lead toxicity in CH_3_NH_3_PbI_3_ perovskites limits their wide application in ultra-wideband photodetectors. To address this issue, Gu et al. [65] prepared a PTE-based detector using a lead-free Cs_3_Cu_2_I_5_ nanolayer film over a large area using a dual-source coevaporation technique. The PTE detector exhibited self-powered optical response wavelengths from the visible (532 nm) to terahertz (119 μm) range. Under 532 nm and 119 μm laser irradiation, the maximum responsivity of the device was 49.2 mA W^−1^ and 3.7 mA W^−1^, respectively. The bandwidth of photodetectors can be extended by utilizing composite materials that combine different components with complementary properties. For example, combining organic and inorganic materials or combining nanostructured materials within a photodetector can achieve a broadband response. Li et al. [66] developed high-performance, self-powered, and flexible PTE-photodetectors based on laser-scribed reduced graphene oxide (LSG)/CsPbBr_3_. Comparative experiments with LSG PDs and basic electrical properties showed that the LSG/CsPbBr_3_ devices exhibited enhanced ultra-wideband photodetectivity at zero bias. Figure 6c presents the schematic structure, mechanism schematic, and photoelectric performance of the LSG/CsPbBr_3_ PD. The PTE potential difference across the device is due to the transport of hot carriers when the laser is irradiated on one side of the electrode. The device covers the UV to terahertz range, with a response of 100 mA W^−1^ and 10 mA W^−1^ for 405 nm and 118 μm, respectively. The maximum *D** of the device is 1.6 × 10^11^ Jones under 532 nm illumination. In the same year, the same research group discovered an organo-inorganic halide chalcogenide CH_3_NH_3_PbI_3_ (MAPbI_3_) and poly(3,4-ethylenedioxythiophene): poly(4-styrenesulfonate) (PEDOT: PSS) [67] with low thermal conductivity, a high Seebeck coefficient, and high carrier mobility composite. An increase of an order of magnitude in the Seebeck coefficient was observed, which was attributed to the addition of PEDOT: PSS. The device showed a stable and reproducible photoresponse at room temperature at 1064 nm and 2.52 THz illumination at zero bias. The temperature variation in the device under the above two kinds of light illumination was measured using an FLIR infrared imaging device, and the results are shown in Figure 6d. Although the performance of MAPbI_3_/PEDOT: PSS PD needs to be improved, it still provides a new avenue for the construction of NIR–THz broadband detectors.

**Figure 6 micromachines-15-00427-f006:**
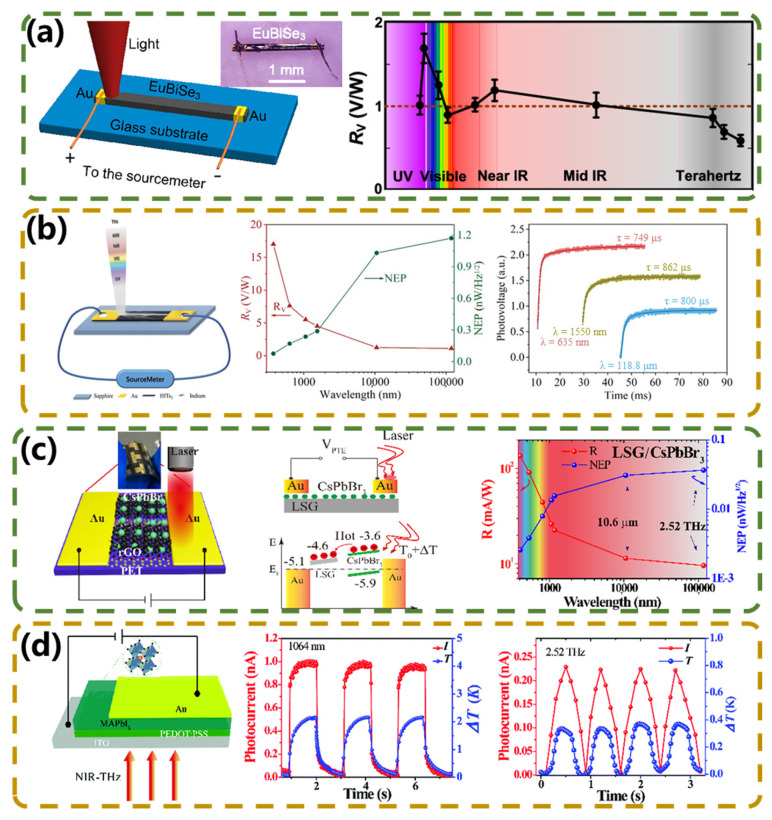
PTE−type broadband photodetectors based on other materials. (**a**) Schematic of the EuBiSe_3_–metal contact device and the voltage response with spectral variation. Reproduced with permission from ref. [63], Copyright © 2019 American Chemical Society. (**b**) Schematic of the HfTe_5_−metal contact device, *R_V_* and NEP at different wavelengths, and rise times of crystals. Reproduced with permission from ref. [64], Copyright © 2020 WILEY−VCH Verlag GmbH & Co. KGaA, Weinheim. (**c**) Structure and mechanism schematic of the LSG/CsPbBr_3_ PD, and the ultra−broadband *R* and NEP curves of the device with the wavelength range from UV to THz. Reproduced from ref. [66] under the Creative Common license CC BY. (**d**) Schematic illustration of MAPbI_3_/PEDOT: PSS PD and photocurrent and Δ*T* of the device under 1064 nm and 2.52 THz laser irradiation at 0.8 mW. Reproduced from ref. [67] under the Creative Common license CC BY−NC.

Both photo-thermoelectric and photo-bolometric devices are photodetectors based on the principle of heat detection, and both can achieve broadband response at room temperature. However, photo-thermoelectric devices differ in that they are self-driven and can operate at zero bias. Meanwhile, a technical problem facing the use of photodetectors based on the bolometric effect is the obvious dark current and low signal-to-noise ratio at room temperature with DC bias. The photon-induced thermoelectric effect of PTE photodetectors at zero bias can effectively reduce the dark current by suppressing the 1/f noise and scattering grain noise in the circuit. PTE photodetectors are currently facing the following challenge: photo-thermoelectric conversion involves multiple energy conversion processes from light energy to heat energy and then to electrical energy, in which there is a certain amount of energy loss and the conversion efficiency needs to be improved. For some applications, such as high-speed communications and fast optical measurements, the current photo-thermoelectric effect photodetectors still have room for improvement in terms of sensitivity and response speed. More-active materials with large Seebeck coefficients should be developed to generate high photovoltage responsivity; meanwhile, the speed of detectors needs to be faster to address this issue.

## 4. Broadband Photodetectors Based on Photoelectric Effect

The difficulty in realizing high-speed detection in thermal detectors has limited their application range and value. Broadband photodetectors based on the photoelectric effect utilize the absorption of photon energy to excite the electrons in the material, resulting in the generation of electron–hole pairs, which in turn generate current or voltage signals for the detection of optical signals. Photon-based detectors have an adjustable response range, good signal-to-noise ratio, and very fast response speed. The physical mechanisms of existing broadband photodetectors mainly include the photoconductive effect and the photovoltaic effect. This paper summarizes the recent advances in photon-based photodetectors based on these two effects and photogating effects.

### 4.1. PCE-Type Broadband Photodetector

The operating principle of photoconductive photodetectors is based on the photoconductive effect. When an optical signal is irradiated into a photoconductor, the energy of the photons is absorbed and causes the electrons in the conducting material to jump to the conduction band, generating electron–hole pairs. These carriers lead to an increase in the conductivity of the material. An external voltage is applied to the photoconductive material to control and collect the carriers, thus realizing the process of converting optical signals into electrical signals. Therefore, how to effectively separate the electron–hole pairs is the main research direction to improve photoconductive photodetectors. In recent years, researchers have pursued several major directions in achieving broadband optoelectronic detectors. Firstly, one approach involves designing artificial microstructures to enhance device absorption in specific wavelength bands, thereby achieving a broadband response. The second method focuses on improving the activation/ionization energy of the device’s photoelectric effect. For instance, homojunction interface work function internal photoemission (HIWIP) and Blocked-impurity band (BIB) detectors broaden the spectral response range by optimizing the barrier. Lastly, efforts have been made to explore novel optoelectronic materials with an ultra-broadband spectral response.

In the last few years, quantum well photodetectors (QWPs) have proven to be excellent photon-based IR/THz photodetectors, benefiting from fast response speeds, high sensitivity, and a flexible and tunable photon response range. Researchers have conducted a lot of work to improve the detectivity and broaden the response range of IR/THz QWPs. Ting et al. [68] theoretically analyzed the possibility of detecting incident far-infrared/terahertz radiation using a dopant-assisted in-band absorption mechanism in a quantum well and proposed the concept of a quantum well intra-sub-band photodetector (QWISP). Dai et al. [69] fabricated In(Ga)As quantum ring infrared photodetectors (QRIPs) with a response range from 3 to 100 THz. In recent years, GeSn/SiGeSn quantum well infrared photodetectors (QWIPs) were theoretically analyzed by Ghosh et al. [70]. The results showed that for a particular choice of material and number of traps, the detector can achieve high responsivity in the mid-infrared terahertz region. To broaden the response further, researchers have introduced the plasmonic resonance effect into the detectors. Cheng et al. [71] designed a hybrid antenna consisting of a patch cavity and a metal grating, as shown in Figure 7a. This antenna can effectively localize and enhance the localized electric field strength inside a QWP. The optical properties of the designed antenna were theoretically investigated, and it was found that the electric field can be increased by a factor of about 10^4^ in the infrared region (6–10 μm) and by a factor of about 10^5^ in the terahertz (THz) region (100 μm). These enhancements can greatly improve the performance of the QWP. The operating wavelength of the detector varies with the size of the structure and ranges from the infrared to the terahertz region.

Furthermore, the plasmonic resonance effect has also played a crucial role in III–V-family detectors. Indium antimonide (InSb) has a bandgap of ~0.17 eV at room temperature, which allows it to be used for infrared wave detection via intrinsic absorption. Well-grown, high-quality InSb has high electron mobility and low effective electron mass. According to the Drude model, these properties give it a negative dielectric constant in the millimeter/terahertz wave range, which can be used to help stimulate intense light–matter interactions by introducing surface plasmon polaritons (SPPs). Thus, InSb can be an important material for the development of SPP principle devices in the terahertz wave range. As shown in Figure 7b, Tong et al. [72] reported GaAs epitaxial indium antimonide for infrared and millimeter/terahertz multiband optical detection. For infrared waves, the light response originates from the interband transition of optoelectronic semiconductors; for millimeter/terahertz waves, the light response originates from the nonequilibrium electrons induced by surface plasmon polarons. The detector has a strong response to infrared waves with a cutoff wavelength of 6.85 microns, and the blackbody detection rate at room temperature is 1.8 × 10^9^ Jones. For millimeter/terahertz waves, the detector exhibits broadband detection performance ranging from 0.032 THz (9.4 mm) to 0.330 THz (0.9 mm).

QWPs cannot be excited by perpendicularly incident light due to the limitations of the sub-band-leap selection rule and require the design of additional optically coupled structures to realize IR-THz photodetection. In addition, the extremely low operating temperature of QWPs poses a great challenge for the integration of QWPs and readout circuits (ROICs) due to the failure of ROICs at low temperatures [73]. This factor has led to IR-THz QWP focal plane arrays (FPAs) not being realized internationally so far. To solve the above problem, Bai et al. [74] developed a high responsivity p-GaAs homojunction interface work function internal photoemission (HIWIP) detector. Figure 7c shows the schematic structure and detection mechanism of the detector. The free carrier absorption of IR radiation occurs in the highly doped emitter layer, followed by photoexcited carriers in the internal photoemission. The photoexcited carriers then cross the intrinsic potential barrier and are collected in the presence of an electric field. The test results showed that the response of the optimized detector is significantly enhanced below the Reststrahlen band compared to that of conventional detectors. The peak responsivity of the detector with the bottom reflector was up to 6.8 A W^−1^ at a bias voltage of 1 V. The equivalent noise power was 2.3 pW Hz^−1/2^. By further optimizing the device, the group, for the first time, theoretically a novel terahertz imaging device utilizing HIWIP-LED upconversion [75]. The generated photogenerated carriers are injected into the LEDs of the integrated epitaxial, driving the LEDs to emit near-infrared photons, which are received and then directly imaged using a mature Si CCD. One significant advantage of this pixel-free imaging is its independence from ROIC requirements. Consequently, it inherently addresses the issue of ROIC failure at low temperatures. The device structure, the photocurrent spectra, and the responsivity of the device at different bias voltages are shown in Figure 7d. The imaging device does not require any optically coupled structural design, has a wide spectral detection range (4–20 THz) and a fast response speed (~ps order), and the optimized quantum well light-emitting diode has an external quantum efficiency that is 72.5% higher than that of a conventional double-heterojunction light-emitting diode. With a noise equivalent power as low as 29.1 pW Hz^−1/2^, the device lays the foundation for the development of highly efficient and compact terahertz imaging devices. To further improve the performance of photodetectors in a wide spectral coverage range, Lian et al. [76], from the same research group, theoretically proposed an E-shaped patch antenna with three slots. Due to the combined effect of the antenna and the microcavity, the optical coupling efficiency of the E-patch antenna microcavity HIWIP was improved by a factor of four on average in the wide bandwidth range of 2.1–20 THz compared with the reference HIWIP. With a further etched E-patch antenna microcavity HIWIP that retains only the active region underneath the antenna, the device achieved an average coupling efficiency increase of 15 times in the 2.3–20 THz range.

Blocked-impurity band (BIB) detectors are a special type of extrinsic photon detector with a response band typically in the 5–300 μm range. Due to their high sensitivity, low dark current, and wide response range, BIB detectors have been widely used in astronomical applications. In recent years, a large number of studies have been devoted to the development of BIB detectors, most of which have focused on the improvement in sensitivity. However, the great potential of BIB detectors for ultra-wideband IR/THz detection has not yet been realized. As shown in Figure 7e, Zhu et al. [77] reported an ultra-wideband multiband infrared/terahertz detector based on the principle of blocked impurity band detection. The detector was prepared by injecting phosphorus into germanium (Ge:P). The impurity band reduces the ionization energy of the dopants, thus broadening the response coverage. The peak in the terahertz region (116.5 μm) corresponds to an energy lower than the ionization energy of phosphorus in germanium (Ge), leading to an absorption peak. In the infrared region, this is attributed to the intrinsic defects within the bandgap of Ge. The response spectra of the detectors showed ultra-wide and double response bands in the range of 3–28 μm (infrared band) and 40–165 μm (terahertz band), respectively. However, due to the operational principles of the BIB detector, the working temperature of this detector (4.5 K) is lower than that of most detectors.

**Figure 7 micromachines-15-00427-f007:**
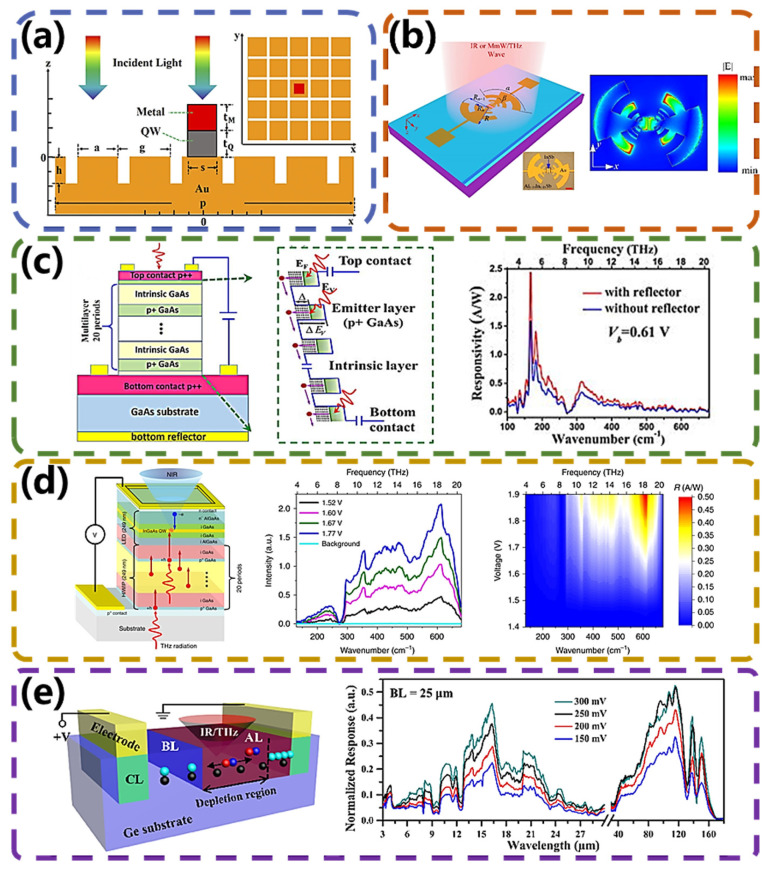
PCE−type broadband photodetectors. (**a**) Schematic of the hybrid antenna. Reproduced from ref. [71] under the Creative Common license CC BY. (**b**) Schematic of a multiband detector, typical field distribution for antenna coupling, and the responsivity and NEP of the detector with respect to temperature. Reproduced from ref. [72] under the Creative Common license CC BY. (**c**) Structure diagram, band diagram, and detection mechanism of p−GaAs HIWIP detector. Reproduced with permission from ref. [74], Copyright © 2018 AIP Publishing. (**d**) Structure of HIWIP−LED upconverter device, photocurrent spectrum, and responsivity under different bias voltages of HIWIP−LED at 3.5 K. Reproduced from ref. [75] under the Creative Common license CC BY. (**e**) Ge:P BIB detector schematic and detector response spectra at different bias voltages. Reproduced from ref. [77] under the Creative Common license CC BY.

Researchers are also developing several new materials with ultra-broad spectral responses. Dielectrics with a very narrow bandgap while maintaining semiconductor-like conductivity have been prioritized for ultra-broadband photodetection. EuSbTe_3_ detectors with a broad spectral response from the ultraviolet to terahertz range at room temperature were reported by Niu et al. [78]. The photoluminescence of the device was greater than 1 A W^−1^ in the full detectable wavelength band, and the NEP was below 1 nW Hz^−1/2^ up to the terahertz wavelength of 163 µm. This two-dimensional layered crystal structure is very advantageous for advanced integration techniques. A two-dimensional photoelectronic material based on a thermal phase transition was also developed by Wu et al. [79]. Unlike ordinary metals, where only single-particle excitations exist, charge density wave (CDW) materials 1T-TaS_2_ have collective excitations called amplitude and phase modes. It is very sensitive to light at room temperature and responds in the visible to terahertz spectral range. Devices using TaS_2_ as the sensitive material have an ultra-broadband optical response with a current responsivity of ~1 A W^−1^ at room temperature. This discovery opens up a new avenue for realizing uncooled, ultra-broadband, and highly sensitive optoelectronic devices in the continuous spectral range from the visible to the terahertz range.

In addition to the three methods mentioned above, there exists a novel way to broaden the band of PCE-type photodetectors. HgCdTe detectors, being one of the most established commercial detection technologies, have demonstrated an expanded detection range through a reduction in operating temperature. Wehmeier et al. [80] explored the utilization of liquid-helium-cooled Hg_1−X_Cd_X_Te (MCT) as a rapid detector for near-field nanospectroscopy. The liquid-helium cooling enabled the MCT to perform unprecedented broadband nanospectroscopy, spanning from 5 to 50 THz, while exhibiting a speed of less than 10 ns compared to the common T = 77 K operation, having a profound impact on the near-field technique.

In recent years, research advances in broadband photoconductive photodetectors have focused on the use of detector structures with multilayer stacking of different materials and the selection of detector materials with bandgap structures covering a wider range of wavelength bands. The biggest advantage of photoconductive detectors is the fast response speed, which can meet the needs of high-speed detection applications. Future photoconductive ultra-wideband photodetectors have to overcome the problem of their significant response performance differences in different bands.

### 4.2. PVE-Type Broadband Photodetectors

For photovoltaic detectors, when an optical signal hits a photovoltaic photodetector, the energy of the photons is absorbed, causing the electrons in the conducting material to jump to the conduction band, forming free electrons and creating holes. These photogenerated carriers start moving under the action of the built-in electric field, forming a photogenerated current. By connecting a photovoltaic-type photodetector to an external circuit, changes in the photogenerated current can be measured and recorded, thus enabling the process of converting an optical signal into an electrical signal. Common photovoltaic-type photodetector structures are p–n junctions, heterojunctions, and Schottky junctions.

The spectral response of common photodetectors is limited by the cutoff wavelength limit λ_c_, which is related to the activation energy (Δ) of the semiconductor material through the relationship *λ_c_ = hc/*Δ. This spectral rule dominates device design and inherently limits the long wavelength response of semiconductor photodetectors. A novel photodetector, named optical pumped hot hole effect detector (OPHED), which overcomes spectral limitations, was reported by Lao et al. [81]. The detector is based on a hot- and cold-hole energy-transfer mechanism. Hot carriers injected into the semiconductor structure interact with cold carriers and excite them to higher energy states. Subsequently, based on this theory, Bai et al. proposed an ultra-broad-spectrum photon-type detector based on a new GaAs/AlxGa_1−x_As quantum ratchet structure [82]. As shown in Figure 8a, the detector was capable of realizing a positive-incidence response with a response range covering 4–300 THz, which far exceeds the coverage of other photon-type detectors. In the infrared wavelength range, absorption is based on the rattling-type valence band, and the generated charge carriers are swept out and collected by the electrodes. Due to the photon energy exceeding the barrier energy, infrared absorption can be achieved at low bias voltages. In contrast, terahertz energy is much smaller than the barrier energy. The response mechanism in this wavelength range involves the process of cold–hot hole energy transfer. When a negative bias voltage is applied to the device, a significantly tilted and curved gradient barrier allows for the injection of a large number of hot holes into the absorber. Excess-energy hot holes undergo inelastic scattering with cold holes below the hole quasi-Fermi level, rapidly reconstructing a quasi-steady-state distribution of thermally excited hot holes. This distribution can reduce the activation energy of charge carriers in the absorber, thereby enabling a terahertz response. The device generated significant photocurrents even at zero bias voltage, with a peak responsivity of 7.3 A W^−1^, which is five orders of magnitude higher than that of OPHED. Due to the asymmetry of the quantum ratchet energy band structure, the device response also exhibited significant differences at positive and negative bias voltages. At temperatures below 77 K, the detector exhibited significant rectification behavior due to the quantum ratchet effect, and the device’s dark current was much lower than that of existing photon-type detectors, with a noise equivalent power as low as 3.5 pW·Hz^−1/2^ and a detection rate as high as 2.9 × 10^10^ Jones, demonstrating its potential for operation at high temperatures. Subsequently, they developed an ultra-wideband upconversion device by directly connecting a p-type GaAs/AlxGa_1−x_As ratchet photodetector (RP) in series with a GaAs double-heterojunction light-emitting diode (DH-LED) using molecular-beam epitaxy based on this structure [83], as shown in Figure 8b. The ultra-broadband optical response of this upconversion device from the terahertz (THz) to the near-infrared (NIR) region (4–200 THz) covers a wider frequency range than other upconversion devices. Broadband IR/THz radiation from a 1000 K blackbody was upconverted to NIR light and detected using a commercial silicon-based device.

Compared to reducing the activation energy of charge carriers, developing narrow-bandgap materials to construct novel heterostructures represents an alternative approach for extending the operational spectrum of photodetectors. Theoretical calculations have shown that opening and changing the bandgap in graphene can extend the operating spectrum of graphene photodetectors into the far-infrared (FIR) and even terahertz (THz) regions. However, the fabrication process to create the bandgap in graphene is complex. Recent studies have shown that graphene oxide (RGO) has a natural energy gap, and therefore, RGO has been widely used in the study of MIR/THz photodetectors. In theory, the RGO energy gap is not suitable for visible and near-infrared light detection. However, Si nanowire (SiNW) arrays have two unique advantages that can compensate for the shortcomings of RGO to some extent. One is the Si bandgap of ~1.12 eV, which makes SiNW arrays suitable for light detection in the visible to NIR range. The other is the aligned nanowire array structure, which can strongly suppress light reflection over a wide spectral range from visible to NIR. Cao et al. [84] integrated the advantages of RGO and SiNW arrays in a single photodetector in 2014. As shown in Figure 8c, by simply dropping a suspension of GO nanosheets on top of the SiNW arrays followed by heat treatment, an RGO-SiNW array heterojunction was formed. The photodetector was photoresponsive to VIS (532 nm), NIR (1064 nm), MIR (10.6 μm), and 2.52 THz (118.8 μm) radiation at room temperature. To the best of our knowledge, this range of photoactivity from the visible (532 nm) to terahertz (118.8 μm) range is the widest range reported so far for graphene-based photodetectors. Due to their Dirac-like surface states, topological insulators (TIs) have also been used to fabricate broadband photodetectors from infrared to terahertz. Yao et al. [85] reported an ultra-wideband photodetector based on a vertically constructed topological insulator Bi_2_Te_3_-Si heterostructure. Figure 8d depicts the structure and schematic diagram of the device. When a Bi_2_Te_3_ film is deposited onto a silicon substrate, the electrons in the silicon move toward Bi_2_Te_3_ to keep their Fermi energy levels aligned with each other, resulting in an accumulation of electrons in Bi_2_Te_3_ and holes in the silicon. In thermal equilibrium, an internal electric field directed from Si to Bi_2_Te_3_ is formed. When light irradiates the device, the corresponding optically active layer absorbs the incident photons and generates electron–hole pairs. The photogenerated carriers are rapidly separated by the built-in strong electric field and finally collected by the electrodes, thus generating a photocurrent in the external circuit. The device demonstrated room-temperature photodetection from UV (370.6 nm) to terahertz (118 μm) with good reproducibility. The photons in the UV to visible-light band are absorbed by silicon, photons in the IR band are absorbed by the bulk states of Bi_2_Te_3_, and for the terahertz band, the surface states of Bi_2_Te_3_ induce the generation of photocurrents. Under bias conditions, the visible light responsivity reached ca. 1 A W^−1^ with a response time better than 100 ms. As a self-powered photodetector, it exhibited very high photosensitivity close to 7.5 × 10^5^ cm^2^ W^−1^ and up to 2.5 × 10^11^ Jones.

Designing materials and structures to achieve efficient light absorption and the separation and collection of electron–hole pairs is a research challenge in achieving high-efficiency photovoltaic broadband photodetectors. This paper concludes that detectors with vertical structures can be fabricated, and quantum effects can be utilized to enhance the performance and response speed of the detectors. In the future, the integration of various materials can also be employed to achieve broadband responses. The self-powering characteristics of photovoltaic photodetectors allow for the creation of more complex, low-power systems, such as imaging instruments and communication devices.

### 4.3. PGE-Type Broadband Photodetectors

The photogating effect is a special type of photoelectric effect, which operates through the photoinduced trapping of electric charges rather than the generation of a photocurrent by carrier excitation [86]. It is typically associated with the presence of trap states within a material that is capable of trapping the excited electrons and holes generated by incident light, thereby altering the potential energy at the semiconductor/dielectric interface. These alterations in charge distribution can induce additional electric gating fields, resulting in shifts in the threshold voltage (V_TH_). In recent years, most of the research on PGE-type photodetectors has been focused on low-dimensional materials such as graphene, transition metal dichalcogenides, and carbon nanotubes. The reported detectors have also been expanded, but only to the infrared band. This section aims to provide a concise overview of the recent advances in broadband photodetectors based on the photogating effect.

Graphene serves as a prominent active channel material in inducing photogating effects within photodetectors [87,88,89]. Nevertheless, the intrinsic low optical absorption of monolayer graphene leads to diminished quantum efficiency and responsivity in conventional graphene-based photodetectors. The integration of colloidal quantum dots in the light-absorbing layer has shown promise in enhancing the responsivity of graphene photodetectors to 1 × 10^7^ A W^−1^ [90]. Since light absorption occurs in quantum dots, the spectral range of the photodetection is small. However, this enhancement comes at the expense of a narrower spectral range for photodetection, as light absorption predominantly occurs within the quantum dots. In 2014, Liu et al. [91] reported an ultra-broadband photodetector based on a double-layer heterostructure of graphene. Under optical illumination, photoexcited hot carriers originating from the top layer tunnel into the bottom layer, inducing charge buildup on the gate and thereby manifesting a strong photogating effect on the channel conductance. The device achieved room-temperature photodetection capabilities spanning from the visible to mid-infrared spectrum, boasting a mid-infrared responsivity exceeding 1 A W^−1^, which holds promise for the development of graphene-based hot-carrier optoelectronic applications.

However, due to the need for external bias, the detector suffers from large dark currents and considerable noise. To address this issue, Gao et al. [92] developed a zero-bias graphene-based photogating effect detector by tuning the asymmetric structure of the graphene channel with cadmium sulfide (CdS) films. Graphene coated with CdS nanocrystals acted as a p-doped channel. Upon illumination, photogenerated carriers are effectively separated within the CdS nanocrystals. When holes are trapped in CdS, the electrons are injected into the graphene, and the Fermi energy level of the graphene increases, leading to the flow of photocurrent (I_Ph_). A notable visible-light responsivity of 0.26 A W^−1^ was achieved through this method. Despite the prevalent use of silicon (Si) as a substrate, its limited cutoff wavelength confines the detection capabilities of optoelectronic devices within the visible to near-infrared (NIR) range [93]. To circumvent this limitation, Fukushima et al. [87] prepared mid-wavelength infrared (MWIR, 3–5 µm) photoelectronic devices, which can be used in the industrial and aerospace industries, by replacing the Si substrate with indium antimonide (InSb) as the active layer.

Recent research efforts on devices based on the photogating effect have predominantly centered around two-dimensional materials. Consequently, there is a pressing need to explore a broader spectrum of active materials. Moreover, in terms of detection range, there remains a necessity to devise methodologies for broadening the spectral coverage and extending detection capabilities into the terahertz or even microwave bands.

## 5. Broadband Photodetectors Based on Other Effects

### 5.1. Broadband Photodetectors Based on Nonlinear Hall Effect

In 2019, He et al. reported a novel nonlinear planar Hall effect (NLHE) observed in 3D nonmagnetic topological insulators [94]. Unlike the linear Hall effect, the nonlinear effect has time-reversal symmetry and is a second-order electron transport phenomenon that can induce frequency doubling. It exhibits a voltage with a frequency component twice that of the driving alternating current (AC) and a steady component that is generated owing to the rectification effect, transforming the AC signal into a direct current (DC) signal. Consequently, it has potential applications in various fields requiring second-harmonic generation or rectification, including energy harvesting, wireless communications, and infrared detectors [95]. Isobe et al. [96] substantiated these findings through theoretical studies, elucidating the potential of the nonlinear Hall effect for high-frequency rectification. Furthermore, Kumar et al. [97] observed the nonlinear Hall effect in Weyl semimetal TaIrTe_4_ at room temperature. Their research achieved 2.4 GHz wireless RF rectification under zero external bias and magnetic field conditions, underscoring the practical significance and versatility of this effect. Subsequently, researchers have explored the application of the nonlinear Hall effect in the terahertz range at room temperature. Hu et al. [98] reported the observation of the second-order NLHE in type-II Dirac semimetal CoTe_2_ under time-reversal symmetry conditions. This phenomenon enables room-temperature terahertz rectification. They revealed remarkable photoresponsivity exceeding 0.1 A W^−1^, response times of approximately 710 ns, and an average NEP of 1 pW Hz^−1/2^ within the 0.1–0.3 THz range. This discovery paved the way for novel approaches to low-energy photon harvesting through NLHE-induced nonlinear rectification, showing significant potential in the field of infrared/terahertz photonics. Meanwhile, He et al. [99] achieved the selective growth of type-II Weyl semimetal NbIrTe_4_ using a self-flux method. This semimetal features strongly tilted Weyl cones and exotic Fermi arcs. By engineering the antenna-induced oscillating terahertz field through a planar metal–topological semimetal–metal structure and van der Waals stacking, they achieved self-powered photodetection at room temperature. The results showed the photodetector achieved a high responsivity of up to 264.6 V W^−1^ at 0.3 THz, fast response times of up to 1 µs, and a low NEP of < 0.28 n W Hz^−1/2^.

Theoretically, the NLHE has the potential to broaden the optical response of detectors extending into the far-infrared region, but there are few examples of realized infrared responses. Two distinct approaches have been used to realize the response to infrared light in the current research. Zhang et al. [100] discovered the potential of new materials such as the Weyl semimetal NbP and the ferroelectric semiconductor GeTe for terahertz/infrared photodetection. Their findings revealed that these materials allow detection up to 15 THz, boasting a remarkable current responsivity of 3 A W^−1^. At higher frequencies, the interband linear photogalvanic effect (LPGE) emerges as a dominant factor [101]. The large current responsivity in both NbP and GeTe opens up the possibility of expanding detection to infrared light. Additionally, different properties of the same material are also worth investigating. Sun et al. [102,103] have detected broadband photocurrents spanning the hundreds of terahertz range by leveraging the third-order response surviving in bulk TaIrTe_4_, while Kumar et al. [97] achieved radiofrequency rectification using the second-order response contributed by the NLHE on the surface of TaIrTe_4_. These findings indicate the vast untapped potential within the same material across a broad infrared–terahertz spectrum, which is poised to capture the interest of researchers.

### 5.2. Broadband Photodetectors Based on the Combination of Multiple Effects

Some broadband IR/THz photodetectors based on the combination of multiple detection mechanisms have been developed in recent years. So far, significant progress has been made in the field of perovskite-based high-sensitivity photodetectors, and several attempts have been made to extend their response wavelength from the ultraviolet to the visible range. However, the detection wavelengths are still less than 2 μm, not reaching the mid-infrared or terahertz bands. Researchers have found that thermoelectricity is another important property of perovskite, which is a promising thermoelectric material with ultra-low thermal conductivity and a high Seebeck coefficient. Since there is no wavelength selectivity, these types of PDs always have a large range of responses. Therefore, perovskite thermoelectric PDs are expected to extend the photoresponsive wavelength range to the THz band and will be widely used in the future. Li et al. [104] prepared a dual-mechanism CH_3_NH_3_PbI_3_ (MAPbI_3_) PD, which combines the photoconducting response in the UV–VIS range and the bolometric response in the MIR–THz range. Figure 9a shows a schematic structure of the device and the optoelectronic response with wavelength. The results show that the MAPbI_3_ PDs, due to their optoelectronic–thermoelectric dual-mode operation mechanism, could realize an optoelectronic response covering the UV to THz range with ultra-broadband photodetection, with high responsivities of 10^5^ mA W^−1^ and 10^2^ mA W^−1^, respectively. In addition, a fast response time of 76 ns was measured using a 1064 nm pulsed laser. This work opens the door for the development of chalcogenide-based thermoelectric detectors and demonstrates that MAPbI_3_ is a promising material for ultra-broadband PD.

Many graphene photodetectors were mentioned above, but the absorption of light by a single layer of graphene is only 2.3%, resulting in a very low responsivity (~6.3 mA W^−1^) of 2D GFET photodetectors. Although combining graphene with a photosensitive substance can substantially improve the responsivity of a photoelectric sensor, the bandwidth and responsivity are severely impaired. Therefore, Deng et al. [105] innovatively proposed a method of using a silicon nitride stress layer to drive 2D GFETs to self-curl into microtubular 3D GFETs, and, for the first time, arrays of 3D GFET devices with an accurately controllable number of curling turns and radius were fabricated. This 3D GFET can be used as a photoelectric sensor with photovoltaic and photovoltaic effects as a synergistic mechanism, and the operating wavelength range extends from the ultraviolet (325 nm) region up to the terahertz (119 μm) region, which is the widest bandwidth of photoelectric sensors based on graphene materials that have been reported to date. As shown in Figure 9b, the GFET is in the shape of a curled microtube, and the simulated electric field amplitude distribution indicates that the incident light resonates near the inner and outer surfaces of the 3D GFET. The performance of this 3D GFET is much higher than that of 2D GFEsT, combining ultra-high responsivity and ultra-fast response, with a responsivity of more than 1 A W^−1^ in the UV in the visible region and up to 0.23 A W^−1^ in the terahertz region, and a response time as fast as 265 ns. In addition to the thermal phase change principle, the pyroelectric effect is often a solution for broadband photodetectors. Fang et al. [106] reported a self-powered ultra-wideband photodetector monolithically integrated on a 0.72Pb (Mg_1/3_Nb_2/3_) O_3_–0.28PbTiO_3_ (PMN–28PT) single crystal, which is shown in Figure 9c. By combining optothermal and thermoelectric effects, the multifunctional PMN-28PT single crystal responds to a wide wavelength range from the ultraviolet to terahertz regions. At room temperature, the photodetector can generate pyroelectric currents under the intermittent irradiation of incident light without external bias. The optical response was systematically investigated. The pyroelectric current is almost linearly related to the light intensity.

### 5.3. Broadband Photodetectors Based on Electromagnetic Induction Wells Effect

In addition to various thermal effects, Ma et al. [107] discovered a new photoconductive effect. They developed an all-photodetector with simultaneous coverage of visible, infrared, terahertz, and millimeter waves. The photosensitive material of the detector is Te, and Figure 10a shows the structure of the Te detector and the detection mechanism in different wavelength bands. In the VIS and IR bands, nonequilibrium carriers are formed due to the energy of the incident light exceeding the band gap of Te (*hv > E_g_*). In the THz and MMW bands, the photon energy is much smaller than the Te bandgap of 0.3 eV, and additional carriers cannot be generated by interband-leaping light. Electromagnetic induction traps and antisymmetric electric fields are formed in the device when the detector is irradiated. During the half cycle in which the antisymmetric electric field appears parallel to the surface of the detector, electrons in the metal electrodes are emitted into the semiconductor Te through the Lorentz force. This effect is known as the electromagnetic induction wells (EIW) effect, based on which the detector also exhibits excellent response in the THz and MMW bands, with response rates of 9.83 A W^−1^ at 0.305 THz, 24.8 A W^−1^ at 0.25 THz, and 87.8 A W^−1^ at 0.172 THz. The synergy of the photoexcited electron–hole pairs and the EIW effect has a significant impact on the broadening of the response band and the improvement in the response speed of the photodetector. The synergistic effect of photoexcited electron–hole pairs and the EIW effect is of great theoretical significance for broadening the response band and improving the response speed of photodetectors. Chen et al. [108], from the same group, also wanted to broaden the detector response band based on this effect, and they focused on atomically thin 2D materials. They reported the construction of antenna-assisted Bi_2_O_2_Se photodetectors to realize multimechanism broadband photodetection in the IR to THz range, with a response of 58 A W^−1^ at 1550 nm and 2.7 × 10^4^ V W^−1^ at 0.17 THz. To validate the practical applicability of Bi_2_O_2_Se photoconductive devices, room-temperature THz single-pixel imaging tests were performed on them. A modulated 0.17 THz wave was used as a THz source and focused on the device by using a series of lenses. A schematic of the Bi_2_O_2_Se device on a Si/SiO_2_ substrate and THz imaging of the metal key is given in Figure 10b. In addition to Bi_2_O_2_Se, the two-dimensional material VSe_2_ is gradually being developed for optoelectronic applications. Qiu et al. [109] used VSe_2_ to prepare room-temperature broadband photodetectors for VIS into the THz region, which had a synergistic multimechanism, producing a photoconductivity effect, a thermal effect, and EIW. As shown in Figure 10c, in the visible–infrared region, the photoresponse was induced by the photothermal effect of photoexcited electron–hole pairs and external bias voltage, with a responsivity of 1.57 A W^−1^ at 635 nm. In the terahertz region, the nonequilibrium carriers were induced by electron injection into metal electrodes, with a responsivity at 0.027 THz of 1.25 × 10^4^ A W^−1^. At the same time, a fast response (~3 μs) and a rather low noise-equivalent power (~2.0 × 10^−14^ W Hz^−1/2^) were achieved. Meanwhile, the environmental stability of the VSe_2_ broadband photodetector was demonstrated by 1-month air exposure.

Silicon, the most important material in semiconductors, dominates the photodetector market. However, due to its bandgap limitation, Si detectors struggle to detect wavelengths above 1100 nm at room temperature with high sensitivity. In recent years, Si-based broadband detectors have mainly been realized by combining two-dimensional materials to make heterojunction detectors, but all of them suffer from problems of instability and low responsivity. Li et al. [110] chose a single silicon layer as the active layer and prepared an ohmic-contacted three-dimensional metal–semiconductor–metal (MSM)-structured silicon detector by using etching and other processes. The detector is based on various detection mechanisms such as BE, PCE, and EIW and is also highly sensitive to both near-infrared (NIR 3 nm) and terahertz light. It achieved a current responsivity of 0.69 A W^−1^ at 1550 nm, a high sensitivity detection of 0.058 pW Hz^−1/2^, and a fast response of 917 ns at 0.269 THz (1115 μm) at room temperature. The researchers also performed room-temperature terahertz imaging, as shown in Figure 10d. The results of this study open a new avenue for high-performance broadband photodetection of near-infrared and terahertz light beyond the cutoff wavelength of Si detectors, and they are expected to be compatible with mature integrated circuits for high-performance broadband detection.

Using a combination of multiple detection mechanisms to broaden the response range of photodetectors is a common strategy. Different detection mechanisms are suitable for different operating wavelength ranges, allowing detectors to capture light signals from a wider spectral range. Specifically, when detectors are compatible with silicon technology, they can be used in integrated circuits and optoelectronic devices. Table 1 summarizes the key parameters of broadband IR/Thz photodetectors based on various detection mechanisms.

## 6. Conclusion and Outlooks

In conclusion, this review paper comprehensively discussed the development and current status of infrared/terahertz broadband photodetectors, emphasizing their importance and potential applications in various fields. Infrared/terahertz broadband photodetectors have emerged as important optoelectronic devices to fulfill the demand for high-sensitivity, high-speed, and broadband detection in modern communications, as well as in the medical, military, and industrial fields. This paper began with an overview of the development of broadband photodetectors, elucidating their origins and highlighting the factors driving their development. This paper then delved into the major research directions and technological advances in the field. It mainly focused on two types of photodetectors based on the photothermal effect and two types of photoelectric effect, each of which exhibits unique advantages and limitations and is expected to be widely used in different scenarios. In addition, key technologies and challenges surrounding ultra-wideband photodetectors were analyzed. Sensitivity, response speed, and stability emerge as key challenges that require urgent attention. In addition, continuous improvements in material selection, structural design, and fabrication processes are necessary to effectively meet the growing application demands.

Looking ahead, the development of infrared/terahertz broadband photodetectors is very promising and involves great development opportunities. As such, we propose some future research directions in the field of broadband photodetection:Explore new materials and large-area growth: We must continue to study new thermal and photoelectric materials, such as two-dimensional materials, topological insulators, organic–inorganic hybrid materials, and perovskites, which are expected to cover a wider range of wavelength bands and improve the performance and sensitivity of broadband photodetectors. To produce cost-effective FPAs ultimately, research into large-area deposition techniques for materials is also necessary, including various scalable deposition techniques such as magnetron sputtering [111], pulsed-laser deposition [112], atomic layer deposition [113], van der Waals growth [114], and others.Enhanced absorption by artificial microstructures: The optical response range of photodetectors can be broadened via the design and preparation of metamaterials and artificial microstructures. This method involves the use of the special properties of microstructures, such as negative refraction, surface equipartition excitation resonance, etc., to achieve selective absorption of and enhancement in optical signals in different wavelength ranges. This enables the efficient detection of optical signals in a wider range of wavelengths.High-speed and low-power solutions: There is a need to further improve the response speed of photodetectors based on thermal effects so that they can operate in a higher frequency range and realize high-speed imaging and high-frequency signal detection [115]. At the same time, the focus should be on developing self-powered, reduce the size of the device to meet the need for low-power, high-speed communications, as well as portable equipment.Integration functions: Firstly, achieving the integration of multiple functions within an ultra-broadband photodetector, such as enhanced contrast polarimetric imaging [116], multiplexing optical communications [117,118], and dynamic encrypt technology [119], will lead to a high degree of device intelligence. Secondly, exploring the integration of a detector with other functions like signal processing and readout modules will enhance its application efficiency. Finally, 3D stacking techniques can be employed to achieve on-chip intelligence and the large-area integration of arrays.Reliability and stability: There is a need to improve the reliability and stability of ultra-wideband photodetectors, especially under extreme environmental conditions such as high temperature and radiation [120,121], which will enhance their usefulness in challenging applications.Low-cost fabrication techniques: After device specifications all meet the design requirements, the difficulty of the fabrication process needs to be considered. There is a need to develop lower-cost and higher-efficiency fabrication techniques to reduce the production cost of ultra-wideband photodetectors and to facilitate their promotion in large-scale applications. The silicon-based standard CMOS process is the mainstream process for modern IC fabrication, which is a highly mature technology and low in cost, so the device fabrication process should preferably match the silicon-based process [122,123].Cross-disciplinary research and diversified applications: Cross-disciplinary research with other fields (e.g., artificial intelligence, quantum technology, etc.) may bring innovations and breakthroughs to the field of broadband photodetection. Expanding the range of applications in biomedicine, environmental monitoring, security, and other fields will unleash the potential of broadband photodetection to address societal challenges and advance the frontiers of science.

In summary, this overview of infrared/terahertz broadband photodetectors not only provides a comprehensive understanding of the current situation but also lays the foundation for future research and innovation. Through interdisciplinary collaboration and technological breakthroughs, ultra-wideband photodetectors will play a key role in advancing technology and enriching human life.

## Figures and Tables

**Figure 1 micromachines-15-00427-f001:**
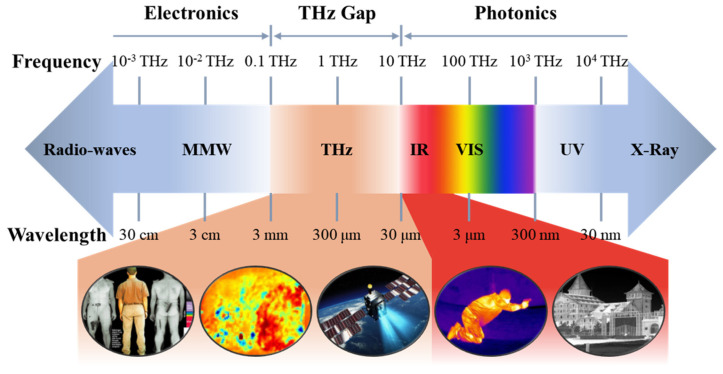
Electromagnetic spectrum and the applications of IR/THz photodetectors. Reproduced with permission from ref. [9] under the Creative Common license CC BY−NC−ND.

**Figure 2 micromachines-15-00427-f002:**
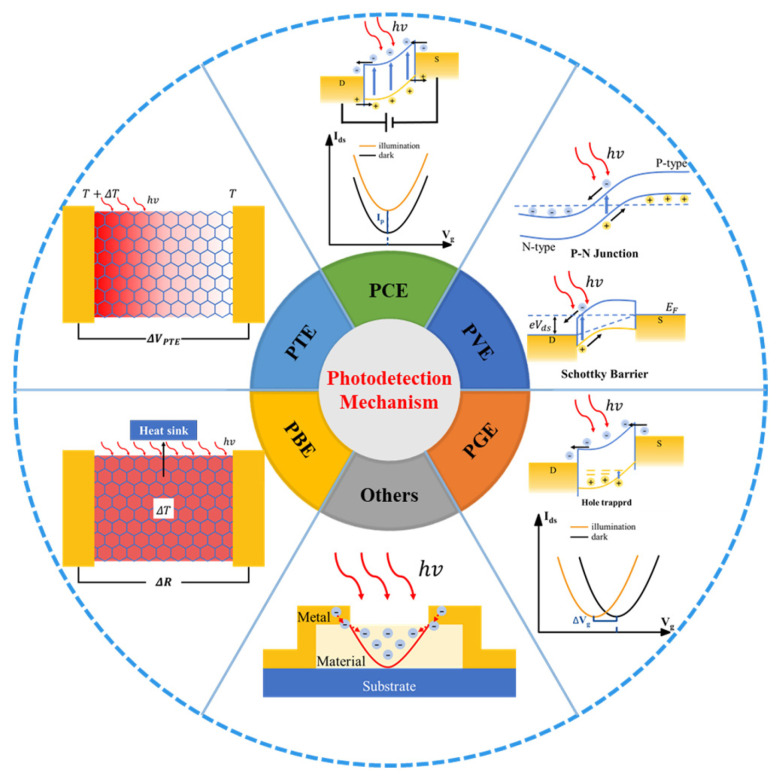
Schematic illustration of photodetection mechanisms. Reproduced with permission from ref. [1], Copyright © 2018 WILEY−VCH Verlag GmbH & Co. KGaA, Weinheim and [3], Copyright © 2021 Wiley−VCH GmbH.

**Figure 3 micromachines-15-00427-f003:**
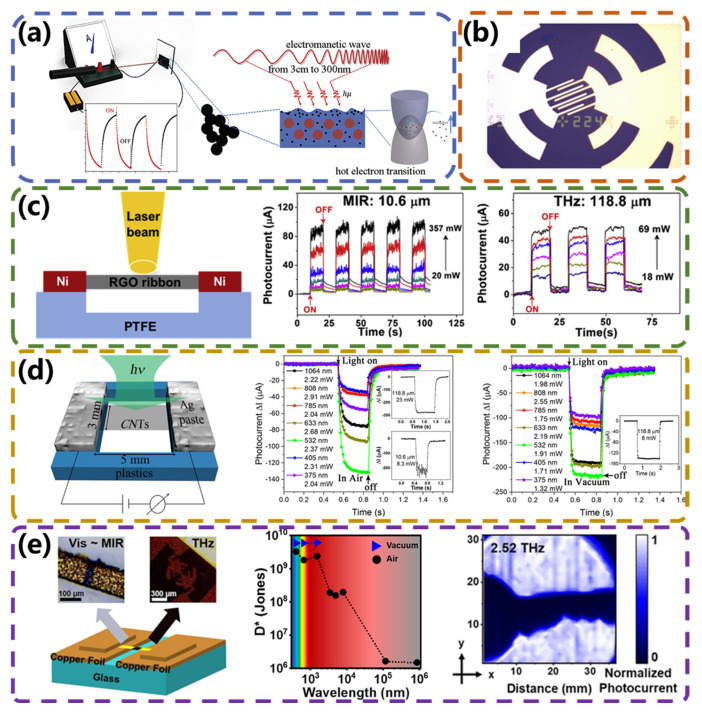
PBE−type broadband photodetectors. (**a**) Test schematic diagram and response principle of CA film. Reproduced with permission from ref. [47], Copyright © 2020 Elsevier Ltd. (**b**) Schematic diagram of logarithmic−periodic antenna. Reproduced with permission from ref. [48], Copyright © 2013 AIP Publishing. (**c**) Schematic and performance of a fully suspended RGO thin film−photodetector. Reproduced with permission from ref. [49], Copyright © 2017 Elsevier Ltd. (**d**) Schematic diagram of the millimetric device structure and the light response curve under different laser irradiation in air and vacuum. Reproduced with permission from ref. [50], Copyright © 2018 American Chemical Society. (**e**) Structural diagram of the VO_2_ (B) device, the multiband optical response, and terahertz imaging results. Reproduced with permission from ref. [51], Copyright © 2022 American Chemical Society.

**Figure 4 micromachines-15-00427-f004:**
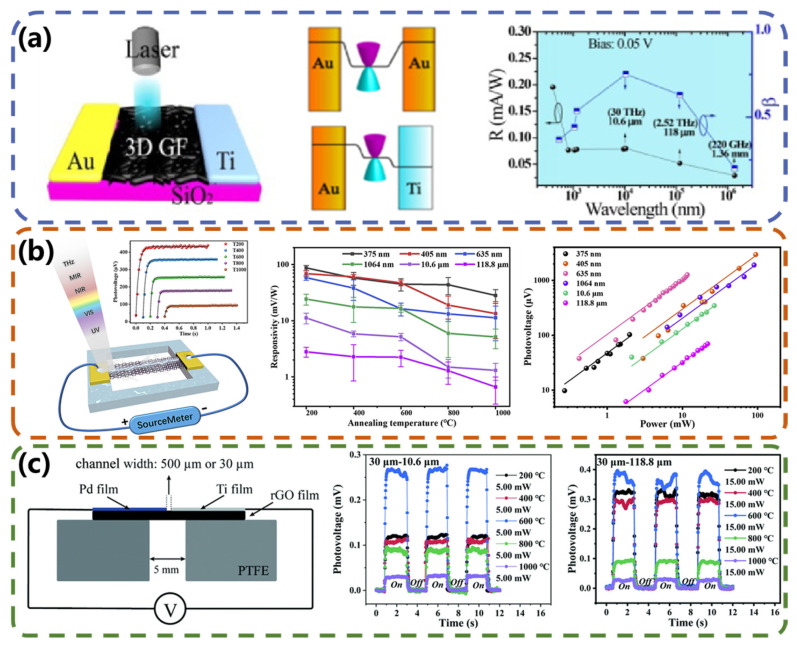
Two-dimensional carbon-based PTE-type broadband photodetectors. (**a**) Schematic structure, band profiles, and the responsivity of the 3D GF photodetector. Reproduced from ref. [53] under the Creative Common license CC BY. (**b**) Schematic, photovoltage responsivities, and power dependence of the photovoltage for the rGO photodetector. Reproduced with permission from ref. [55], Copyright © 2019 Elsevier Ltd. (**c**) Schematic cross-section of the Pd-rGO-Ti photodetector and the photovoltage on–off curve under 2.52 THz illumination. Reproduced from ref. [56] under the Creative Common license CC BY-NC.

**Figure 5 micromachines-15-00427-f005:**
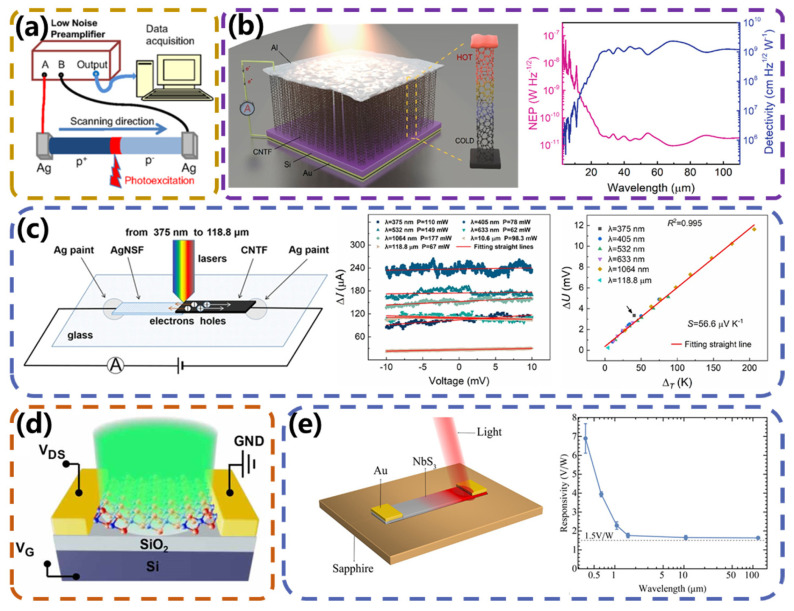
CNT and other two−dimensional materials−based PTE−type broadband photodetectors. (**a**) Schematic of CNT fiber optic photodetector measurement. Reproduced with permission from ref. [57], Copyright © 2018 American Physical Society. (**b**) Schematic of self−powered CNTF photodetector and detectivity as a function of wavelength. Reproduced with permission from ref. [58], Copyright © 2019 American Physical Society. (**c**) Schematic diagram of the heterojunction device and the Δ*I*–*V* curves of their radiated by lasers with wavelengths. Reproduced from ref. [60] under the Creative Common license CC BY. (**d**) A 3D schematic of SnSe_2_−based photodetector. Reproduced with permission from ref. [61], Copyright © 2020 IOP Publishing, Ltd. (**e**) Schematic of NbS_3_−based detector and the photovoltage responsivities in different wavelengths of the photovoltage at room temperature. Reproduced with permission from ref. [62], Copyright © 2020 American Chemical Society.

**Figure 8 micromachines-15-00427-f008:**
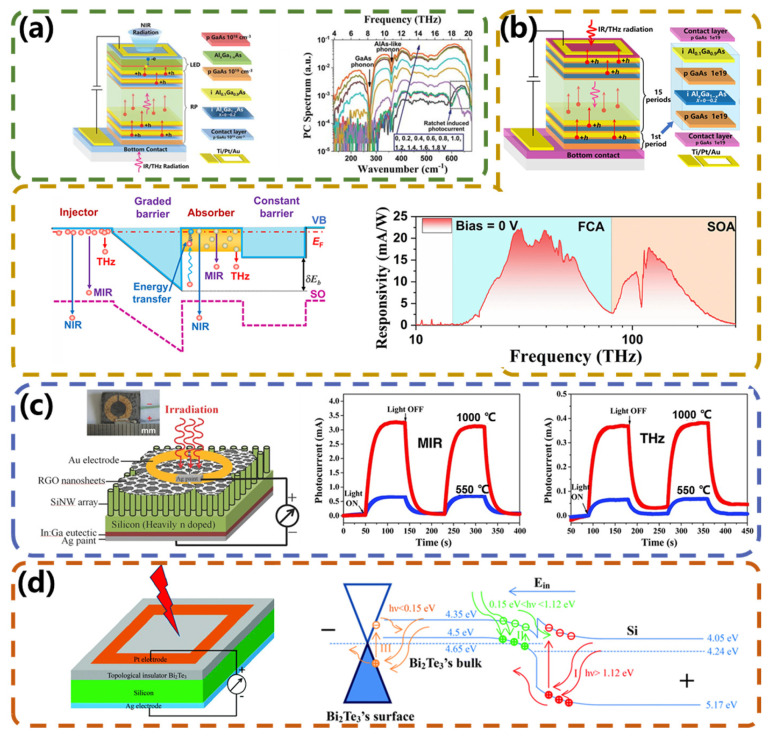
PVE−type broadband photodetectors. (**a**) The structure of a ratchet photodetector, schematic diagram of infrared/terahertz light response, and broadband response achieved by the detector throughout the entire response range. Reproduced with permission from ref. [82], Copyright © 2021 AIP Publishing. (**b**) Structural diagram of RP−LED upconverter and photocurrent spectrum in the range of 4–20 THz at a specific bias voltage. Reproduced from ref. [83] under the Creative Common license CC BY−NC. (**c**) Schematic of the RGO SiNW array and photocurrent response in the infrared/terahertz band. Reproduced with permission from ref. [84], Copyright © 2014 WILEY−VCH Verlag GmbH & Co. KGaA, Weinheim. (**d**) Bi_2_Te_3_−Si heterojunction photodetector and its energy band schematic under reverse source–drain bias. Reproduced with permission from ref. [85], Copyright © 2015 Royal Society of Chemistry.

**Figure 9 micromachines-15-00427-f009:**
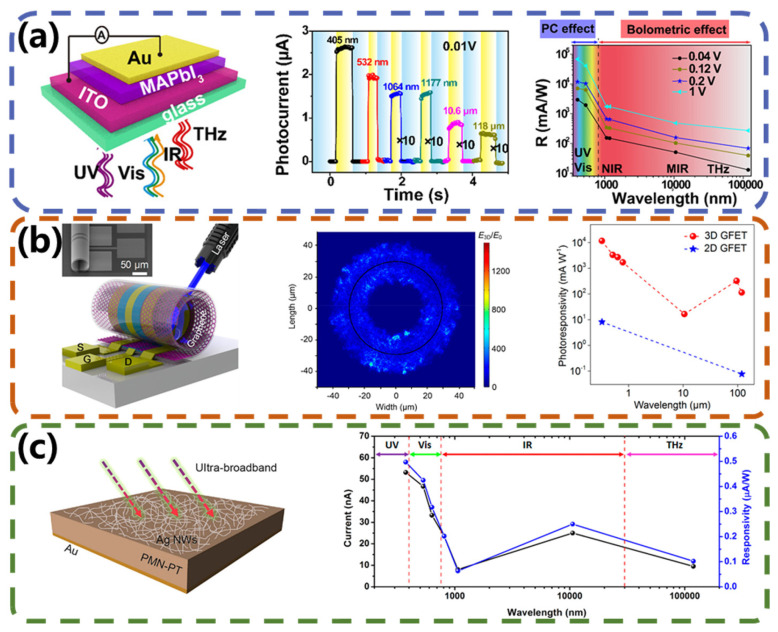
Broadband photodetectors based on multiple mechanisms. (**a**) Schematic structure of the MAPbI_3_ photodetector, ultra−broadband photocurrent in the wavelength range from 400 nm to 118 μm, and full−spectrum photoresponsivity of the MAPbI_3_ photodetector. Reproduced with permission from ref. [104], Copyright © 2020 American Chemical Society. (**b**) Schematic of a 3D GFET, simulated distribution of electric field magnitudes in the vicinity of the device, and comparison of the voltage optical response of a 2D GFET and a 3D GFET. Reproduced from ref. [105] under the Creative Common license CC BY−NC. (**c**) Schematic diagram of an ultra−broadband PMN−PT photodetector and spectral response of the photodetector measured over a wide wavelength range from 375 nm to 118.8 μm. Reproduced with permission from ref. [106], Copyright © 2016 American Chemical Society.

**Figure 10 micromachines-15-00427-f010:**
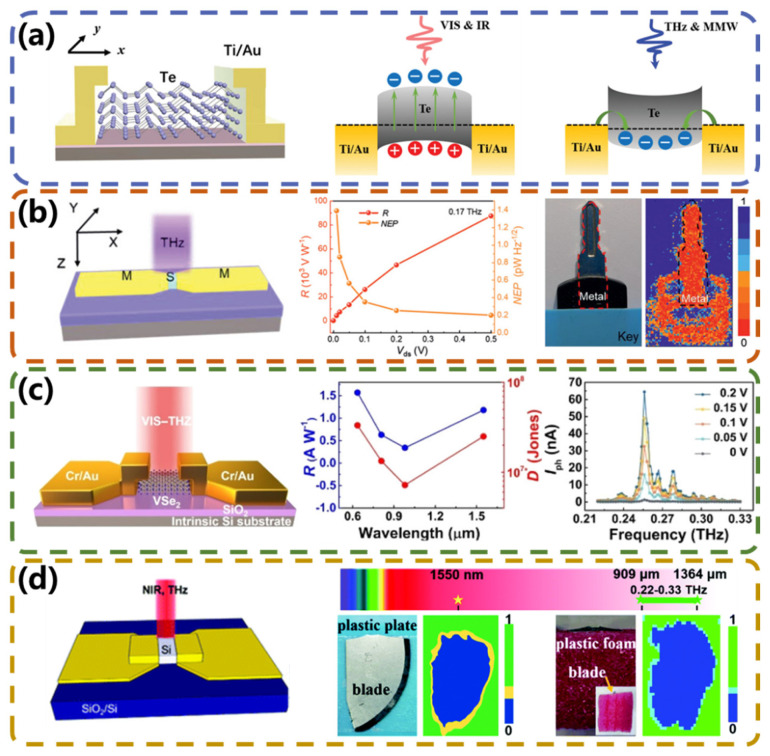
Broadband photodetectors based on the EIW effect. (**a**) Schematic structure of a Te detector and its detection mechanism in different bands. Reproduced from ref. [107] under the Creative Common license CC BY. (**b**) Schematic structure of a Bi_2_O_2_Se terahertz photodetector with symmetric bowtie antenna electrodes, responsivity at 0.17 THz radiation, and imaging of a metal key at THz. Reproduced with permission from ref. [108], Copyright © 2021 Wiley−VCH GmbH. (**c**) Schematic structure of VSe_2_ nanosheet photodetector, *R* and *D** of the device versus incident wavelength, and spectral response of the device at 0.22–0.33 THz frequency at different bias voltages. Reproduced with permission from ref. [109], Copyright © 2020 American Chemical Society. (**d**) Schematic of a 3D MSM−structured Si detector with broadband detection in the electromagnetic spectrum and an image of a blade inside a plastic foam at 0.269 THz. Reproduced with permission from ref. [110], Copyright © 2022 Royal Society of Chemistry.

**Table 1 micromachines-15-00427-t001:** Key figure of merits of some representative broadband IR/THz photodetectors.

Mechanism	Material	Spectral Range	Maximum Responsivity (A W^−1^)	Maximum *D** (Jones) or Minimum NEP	Response Time	Operating Temperature	Bias (V)	Ref.
PBE	Golay	15–8000 μm	1 × 10^5^ V W^−1^	7 × 10^9^	~30 ms	RT *	-	[25]
PBE	Si	15–2000 μm	2.4 × 10^5^ V W^−1^	0.25 pW Hz^−1/2^	-	4.2K	-	[26]
Pyroelectric	LiTaO_3_	0.1–3000 μm	1.5 × 10^5^ V W^−1^	4 × 10^8^	~μs to ms	RT	-	[27]
Pyroelectric	DLaTGS	0.1–>1000 μm	4.1 × 10^6^	2.2 × 10^9^	<100 ms	RT	-	[30]
PBE	CNT	375 nm–118.8 μm	UV−NIR: 0.024 MIR: 0.01 THz: 0.012	UV−NIR: 3.52 × 10^7^ MIR: 1.81 × 10^7^ THz: 2.31 × 10^7^	150 μs	RT	0.2	[50]
PBE	VO_2_(B)	405 nm–880 μm	VIS−NIR: 42.5 MIR: 1.98 THz: 0.038	VIS-NIR: 6.02 × 10^9^ MIR: 1.9 × 10^8^ THz: 1.59 × 10^6^	83 ms	RT	0.2	[51]
PTE	Graphene	1.54 μm–119 μm	IR: 10 V W^−1^ THz: 715V W^−1^	IR: 1100 pW Hz^−1/2^ THz: 16 pW Hz^−1/2^	~30 ps	RT	0.2	[52]
PTE	3D GF	300 nm–1360 μm	10^3^	1 × 10^9^	43 ms	RT	0.05	[53]
PTE	rGO films	375 nm–118.8 μm	NIR: 2.4 × 10^−2^ V W^−1^ MIR: 1.1 × 10^−2^ V W^−1^ THz: 2.8 × 10^−3^ V W^−1^	NIR: 6.5 × 10^5^ MIR: 3 × 10^5^ THz: 7 × 10^4^	34.4 ms	RT	0	[55]
PTE	rGO	375 nm–118.8 μm	1.42 × 10^−1^ V W^−1^	1.58 × 10^7^	100–200 ms	RT	0	[56]
PTE	Ti-CNT-Pd	375 nm–118.8 μm	MIR: 0.35 V W^−1^ THz: 1.27 V W^−1^	MIR: 2.5 × 10^6^ THz: 9 × 10^6^	7 ms	RT	0	[59]
PTE	NbS_3_	375 nm–118.8 μm	UV: 6.9 V W^−1^ VIS: 3.25 V W^−1^ NIR: 1.59 V W^−1^ THz: 1.64 V W^−1^	UV: 1.76 × 10^6^ VIS: 8.3 × 10^5^ NIR: 4.1 × 10^5^ THz: 4.2 × 10^5^	~7 ms	RT	0	[62]
PTE	EuBiSe_3_ crystal	405 nm–118 μm	VIS-NIR: 1.69 V W^−1^ THz: 0.69 V W^−1^	VIS-NIR: 2.9 × 10^8^ THz: 1.2 × 10^8^	207 ms	RT	0	[63]
PTE	HfTe_5_ Crystal	375 nm–118.8 μm	UV: 17 V W^−1^ VIS: 7.6 V W^−1^ NIR: 5.5 V W^−1^ MIR: 1.25 V W^−1^ THz: 1.1 V W^−1^	UV: 2.7 × 10^8^ VIS: 9.3 × 10^7^ NIR: 7.1 × 10^7^ MIR: 1.9 × 10^7^ THz: 1.2 × 10^7^	~1 ms	RT	0	[64]
PTE	Cs_3_Cu_2_I_5_	532 nm–119 μm	VIS: 4.92 × 10^−2^ NIR: 1.1 × 10^−3^ THz: 3.7 × 10^−3^	VIS: 8.2 × 10^7^ NIR: 6.5 × 10^6^ THz: 6.4 × 10^6^	-	RT	0	[65]
PTE	LSG/CsPbBr_3_	405 nm–118 μm	UV: 1.35 × 10^−1^ THz: 1 × 10^−2^	532 nm: 1.6 × 10^10^	18 ms	RT	0	[66]
PTE	MAPbI_3_/ PEDOT: PSS	1064 nm–118 μm	1.6 × 10^−6^	1.2 × 10^7^	28 μs	RT	0	[67]
PCE	GaAs–InSb **	3–6.85 μm and 909–9375 μm	MIR: 5.4 × 10^3^ V W^−1^ THz: 5.6 × 10^4^ V W^−1^	MIR: 1.8 × 10^9^ THz: 0.1 pW Hz^−1/2^	800 ns	RT	1	[72]
PCE	p-GaAs	15 μm–71.4 μm	0.5	29.1 pW Hz^−1/2^	-	3.5K	1.9	[75]
PCE	Ge:P **	3–28 μm and 40–165 μm	MIR: 5 THz: 7.2	MIR: 6.8 × 10^12^ THz: 9.9 × 10^12^	-	4.5K	0.15	[77]
PCE	EuSbTe_3_	532 nm-119 μm	VIS: 8 THz: 1	VIS: 150 pW Hz^−1/2^ THz: 0.9 nW Hz^−1/2^	~8 ms	RT	1.2	[78]
PCE	1T-TaS_2_	532 nm–119 μm	VIS: 3.92 THz: 0.76	VIS: 80 pW Hz^−1/2^ THz: 0.4 nW Hz^−1/2^	~1.5 ns	RT	0.71	[79]
PVE	GaAs/Al_x_Ga_1−x_As	1–75 μm	7.3	2.9 × 10^10^	-	4.2K	0	[82]
PVE	rGO-SiNW	532 nm–118.8 μm	MIR: 9 × 10^−3^	-	10 s	RT	1	[84]
PVE	Bi_2_Te_3_–Si	370 nm–118 μm	635 nm:1	635 nm: 2.5 × 10^11^	<100 ms	RT	5	[85]
PCE, PBE	CH_3_NH_3_PbI_3_	405 nm–118 μm	VIS: 6.8 × 10^2^ MIR: 0.483 THz: 0.271	VIS: 1.2 × 10^9^ MIR: 2.6 × 10^7^ THz: 1.9 × 10^7^	126 ns	RT	1	[104]
PTE, PVE	3D graphene FET	375 nm–119 μm	UV-VIS: >1 THz: 0.23	THz: 2.8 × 10^10^	~265 ns	RT	0	[105]
Optothermal and pyroelectric	PMN–PT	375 nm–118.8 μm	IR: 8.8 × 10^−8^		~ms	RT	0	[106]
PCE, EIWs	Bi_2_O_2_Se	940 nm–15,000 μm	VIS-IR:58 THz: 2.7 × 10^4^ V W^−1^	THz: 0.2 pW Hz^−1/2^	476 ns	RT	0.2	[108]
PBE, PCE, EIWs	VSe_2_ **	635–1550 nm and 0.02−0.04 and 0.22−0.33 THz	635 nm:1.57 1550 nm:1.18 THz:1.25 × 10^4^	THz: 2.7 × 10^10^	~3 μs	RT	0.2	[109]
PBE, PCE, EIWs	Si **	1550 nm and 909–1364 μm	1550 nm:0.69 THz: 4.95 × 10^3^	THz: 0.058 pW Hz^−1/2^	917 ns	RT	0.5	[110]

* RT stands for room temperature. ** The detector does not operate continuously throughout the IR–THz spectral range.

## Data Availability

Not applicable.

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
