# Peer review of "Recent Advances in Broadband Photodetectors from Infrared to Terahertz"

_micromachines, 2024, doi:10.3390/mi15040427_

Round 1

Reviewer 1 Report

Comments and Suggestions for Authors

The article reviews a broad range of modern and state-of-the-art single pixel and array detectors for light, focussed on the infrared (IR) to THz spectral range and including devices with performances up to the ultraviolet range. The emphasis is on ultrabroadband detection across the IR-THz range. The basic operating principles of the most common classes of detectors are discussed along with descriptions of more exotic devices.

General comments:

Overall this is an interesting article, with many examples of different kinds of detectors included, and a good number of references. Many applications would benefit tremendously from high performance detectors in this spectral range so this work is likely to be a useful reference. 

However, there are areas where the paper is weak, or where specific improvements should be made as detailed below. I recommend that the paper is published after major revision. 

Detailed comments:

1. Whilst many different types of detector are presented, along with some basic operating principles and performances, there is not much insight added by the authors here. The sections are organized by type of detector e.g. photoelectric, but the discussion jumps from detector to detector without much consideration of which detector might be better than others for a particular application. For example, a discussion of which devices are good for room-temperature portable sensing applications would be useful. The key factors that influence fast response times in the different detector types would be another interesting discussion point.  

2. The key parameters for all detectors discussed in the review are summarised in Table 1, which is an important part of the paper and needs to be improved. A clear reference to the Table should be made in the text before the devices are described. Considering that this article is reviewing IR-THz sensors it is not useful that the spectral range is crudely tabulated as e.g. "IR-THz" or "UV-THz". This column should be updated to give specific spectral range (e.g. in cm-1 or THz units) for each sensor, allowing different detectors to be evaluated. Some of the detectors do not have continuous operation across the spectral range and it would be useful to include this information in the table (perhaps a * denoting "not continuous" for example). One example is Figure 10c where the device has a broad NIR operation, but nothing shown in the mid-IR and only a very narrow (< 0.1 THz) THz range, yet is still described as "IR-THz".      

The operating temperature of each detector should also be added as a new column to allow comparison with very widely used cryogenically cooled detectors in this range.

2. To put all of the detector performances in context, it would be very useful to add performances of some very common types of commercial IR-THz detectors. The Golay cell is shown in Table 1, but it would be useful to include (cooled) Si Bolometers and room temperature (e.g. pyroelectric DLaTGS) detectors. 

3. Different levels and types of information are given for the different examples discussed in the paper, and this makes it very difficult to compare detectors, or to draw conclusions about different classes of detector. For example in the figures, some show a simple "response to light" on-off curve, whilst others have more detailed detectivity D* data. Can this be made more consistent? Sensitivity numbers only really make sense to compare along with some idea of the noise level (NEP) or through conversion to D*.

4. For a significant number of devices, whilst there is some response across a very large spectral range, it is clear that the performance actually falls off dramatically at one end of the spectral range - to a level which would be far below that of a standard broadband detector. For example in Figure 3e and Figure 5b the detectivity of the devices drops by 3 orders of magnitude over a significant portion of the presented spectral range. Is this really useful broadband performance? In this sense, tabulating only the maximum performances in Table 1 is not very useful for comparing devices. If the maximum performance is outside the mid-IR to THz range then it is quite misleading for this review. It would be more useful to tabulate typical performances across different spectral ranges, or, more simply, maximum and minimum performances across what the authors list as the useful spectral range of the detectors.    

5. Regarding the speed of detectors. The authors need to be consistent about what is regarded as "fast" or "slow" response. 

For example in the PTE section, 100 ms response time is described as "slow", whereas in the same section (line 374, Line 395) response times of 43 ms and 50 ms are described as "short". Really they are very similar and "slow".    

6. There is some description of how different devices were made, and in some cases diagrams showing the device structure. However there is no real discussion about these methods or the benefits/challenges of different architectures. In the introduction it is stated they intend "delve into" fabrication processes as well as other areas, but this does not seem to have been done. 

7. In the photoelectric section, one very recent and relevant detector development (published open access) should be included.  Weheimer et al (ACS Photonics (2023) Vol 10, 4329) discuss an MCT detector with 5-50 THz operation with a response time < 100 ns and high detectivity.  

8. The outlook section (numbered points 1-7) is rather weak. Some of the numbered points are somewhat simplistic and obvious, for example point 1 states that new materials (building on the types discussed here) should be explored and that performances should be improved. Other points may be relevant, but they are areas which have not been discussed significantly at all in the paper. For example improving reliability and stability (point 5) is clearly useful but there is little or no discussion for any of the devices about these issues. The conclusion stability emerges as a "key challenge" is not well evidenced in the text. If it is a key challenge, a specific section discussing those issues would be useful.

9. In the introduction, spectroscopy should be added as a key application area.

10. At the beginning of the introduction on lines 52-66 it is not clear why the specific set of examples has been given. Line 52 introduces devices developed "in recent years" but the first seven references (19-26) are nearly a decade old now. I would expect these devices to have developed significantly since.

11. In Line 137, "radiation" should read "radiation power". 

12. Line 166, define Ep and Eg.

13. Line 208, if photo-sensing different to photo-detection?

14. Line 429, the "detection rates" (cm/Hz^1/2) would depend on the excitation power. Elsewhere in the paper the detection rate is defined as D* (cm/Hz^1/2 /W), commonly know as detectivity.

15. Line 512, the figure shows sensitivity 0.1 V/W, not 1 V/W as stated. 

16. In figure 6d it is not clear what the temperature distributions represent.

17. In the discussion of the device of Bai et al (Line 648 - 677) it is implied that the detector solves the problem of requiring low temperature operation. However the results shown in the figure 7d were all obtained at 3.5 K. The following device of Zhu et al was operated at 4.5 K and this should also be clarified in the text.

18. Line 715, define CDW.

19. Line 767, define OPHED.

Author Response

Dear Reviewer,   The attachment is the responses to your comments.   Yours sincerely, Authors  

Reviewer 2 Report

Comments and Suggestions for Authors

In this manuscript, the authors presented an overview of the latest advances in broadband photodetectors from infrared to terahertz. They introduced broadband photodetectors based on different principles and their research progresses, summarized the techniques and strategies used, the challenges faced, and finally looked forward to the future development of this field. The manuscript is recommended for publication after addressing the following issues.

.

1.      In Section 2, the authors listed a number of parametric formulas, and it is suggested to add the corresponding references.

2.      It is recommended to unite the detection rate unit with either Jones or cm Hz^1/2 W^-1.

3.      The authors classified the photodetectors into 6 categories through basic principles, but the manuscript lacks a detailed description of the development for PGE-type photodetectors, please supplement it.

Author Response

Dear Reviewer,   Thank you for your letter and comments concerning our manuscript entitled “Recent advances in broadband photodetectors from infrared to terahertz” (ID: micromachines-2901941). Those comments are all valuable and very helpful for revising and improving our paper, as well as the important guiding significance to our research. We have studied the comments carefully and have made corrections which we hope meet with approval. Revised portions are marked in red in the revised manuscript. The main corrections in the paper and the responses to your comments are also included below.   Your consideration is highly appreciated.   Yours sincerely, Authors  

Reviewer 3 Report

Comments and Suggestions for Authors

In this manuscript entitled “Recent advances in broadband photodetectors from infrared to terahertz”, Si et al. have summarized the latest research progress in broadband photodetectors operating from infrared to terahertz band based on various physical effects. In the end, the potential future research directions have been proposed. The topic of broadband photodetection is of broad interest to the researchers in the light-sensing field as it enables high integration level of optoelectronic system. The review is well organized and written. However, in the current stage, there are still some issues to be addressed. Therefore, a moderate revision is recommended. Please find the comments below.

1. There have been reviews on the related topic (e.g., Appl. Phys. Rev. 2019, 6, 021316; Adv. Mater. 2021, 33, 2008126; Photonics Res. 2021, 9, 2167). So why is an additional review necessary? The necessity should be discussed in detail in the Introduction section.

2. In the “Fundamentals of photodetectors” section, the definition of other parameters evaluating photodetectors such as responsivity, external quantum efficiency, detectivity, NEP, etc., should be provided. 

3. As shown in the review, many of the previous devices have been constructed based on small nanostructures produced by unscalable methods such as mechanical exfoliation. Therefore, “large-area growth” is undoubtedly an important research direction is the upcoming future. As such, related techniques which have demonstrated the large-area deposition capability (magnetron sputtering (e.g., J. Phys.: Condens. Matter 2023, 35 124002), pulsed-laser deposition (e.g., Adv. Mater. 2023, 35, 2211562), atomic layer deposition (e.g., ACS Appl. Mater. Interfaces 2022, 14, 54034–54043), etc.) should be discussed in Section 6.

4. To date, most of the studies have been focused on sensing the light intensity. However, polarization state is also an important information carried by the light and it can enable multiple functions such as enhanced contrast imaging (e.g., Int. J. Extrem. Manuf. 2021, 3 035201), multi-channel optical communications (e.g., Avd. Photon. Res. 2022, 3, 2200119), etc. However, related studies in the infrared to terahertz band are still rare. Therefore, this is also an important direction deserved to be discussed in Section 6.

5. Some reference information is missed/incorrect. For example, Ref. 2, 13, 25, 26, 60, missed article number/page number. Ref. 83, incorrect journal name. In addition, abbreviations and full journal names have both been used. Please unify. 

Author Response

Dear Reviewers:

Thank you for your letter and comments concerning our manuscript entitled “Recent advances in broadband photodetectors from infrared to terahertz” (ID: micromachines-2901941). Those comments are all valuable and very helpful for revising and improving our paper, as well as the important guiding significance to our research. We have studied the comments carefully and have made corrections which we hope meet with approval. Revised portions are marked in red in the revised manuscript. The main corrections in the paper and the responses to your comments are also included below.

Your consideration is highly appreciated.

Yours sincerely,

Authors

Reviewer 4 Report

Comments and Suggestions for Authors

The authors have provided a comprehensive review for the broadband Thz detectors. This review should draw the attention of the researchers working in the long wavelength detector. I think the manuscript can be accepted after answering several questions.

1. Recently, the nonlinear hall effect has been proposed for Thz detection. Theoretically, it can also be broadband for IR to Thz. However, the author does not include this mechanism in the review.  (Adv.Mater.2023, 35, 2209557; Nature Nanotechnology, 16, 421–425 (2021))

2. In table 1, the author lists the bias for comparison. I don't quite understand why the author thinks this is an important parameter. I would rather to know if all the performance is obtained at room temperature or not. It is challenging to get high-performance IR or Thz detectors at room temperature.

Comments on the Quality of English Language

No

Author Response

(The authors gave the same response as above.)

Round 2

Reviewer 1 Report

Comments and Suggestions for Authors

The authors have made extensive improvements to the manuscript, and have addressed the reviewers concerns. 

Author Response

Dear Reviewers:

Thank you for your letter and comments concerning our manuscript entitled “Recent advances in broadband photodetectors from infrared to terahertz” (ID: micromachines-2901941). Those comments are all valuable and very helpful for revising and improving our paper, as well as the important guiding significance to our research.

The authors thank you again for your contribution to this manuscript!

Yours sincerely,

Authors

Reviewer 3 Report

Comments and Suggestions for Authors

The authors have addressed the comments in a comprehensive way. The manuscript has been greatly improved. Therefore, it can be accepted for publication after a minor revision addressing the following issues.

1. Page 7, “2.2.5. Response time Quantum Efficiency”, is the “Quantum Efficiency” redundant?

2. Page 7, equation (10), the positions of t and t are reversed. Please check.

3. Page 8, line 3, “the response time is”, the first letter should be capital.

Author Response

Dear Reviewers:

Thank you for your letter and comments concerning our manuscript entitled “Recent advances in broadband photodetectors from infrared to terahertz” (ID: micromachines-2901941). Those comments are all valuable and very helpful for revising and improving our paper, as well as the important guiding significance to our research. We have studied the comments carefully and have made corrections which we hope meet with approval. Revised portions are marked in red in the revised manuscript. The main corrections in the paper and the responses to the reviewer’s comments are also included below.

Your consideration is highly appreciated.

Yours sincerely,

Authors

Detailed responses to Reviewer:

The authors would like to thank you for your review. Your comments were important and helpful in improving the manuscript. We have made the corresponding revisions to this paper according to your suggestions. Our responses to your comments are as follows.

Question 1. Page 7, “2.2.5. Response time Quantum Efficiency”, is the “Quantum Efficiency” redundant?

[Response]:

Thank you for your careful review of our manuscript. We appreciate your attention to detail and bringing the redundancy in the section title “2.2.5. Response time Quantum Efficiency” to our attention.

We acknowledge the oversight and confirm that “Quantum Efficiency” is indeed redundant in the title. We have promptly made the correction in Page 7 to ensure clarity and precision in our presentation.

Your feedback has been invaluable in improving the quality of our manuscript, and we are grateful for your contribution.

Question 2. Page 7, equation (10), the positions of t and t are reversed. Please check.

[Response]:

Thank you for your careful review of our manuscript. We appreciate your attention to detail.

You are correct in pointing out the error in equation (10) on page 7, where the positions of “t” and “τ” are reversed. We apologize for this oversight. We have promptly revised the equation to ensure accuracy.

Once again, we thank you for bringing this to our attention, and we are grateful for your valuable feedback.

Question 3. Page 8, line 3, “the response time is”, the first letter should be capital.

[Response]:

Thank you for your careful review and valuable feedback on our manuscript.

Regarding the issue raised about capitalizing the first letter in the phrase “the response time” , we appreciate your attention to detail. We have made the necessary corrections in Page 8, line 3 with proper English grammar conventions.

Your input has been instrumental in improving the clarity and accuracy of our manuscript, and we are grateful for your contribution to the peer review process.

The authors thank you again for your contribution to this manuscript!